# LEVERAGING CHARACTERISTICS OF THE OUTPUT DISTRIBUTION FOR IDENTIFYING ADVERSARIAL AUDIO EXAMPLES

## ABSTRACT

Adversarial attacks can mislead automatic speech recognition (ASR) systems into producing an arbitrary desired output. This is easily achieved by adding imperceptible noise to the audio signal, thus posing a clear security threat. To prevent such attacks, we propose a simple but efficient adversarial example detection strategy applicable to any ASR system that predicts a probability distribution over output tokens in each time step. We measure a set of characteristics of this distribution: the median, maximum, and minimum over the output probabilities, the entropy of the distribution, as well as the Kullback-Leibler and the Jensen-Shannon divergence with respect to the distributions of the subsequent time step. Then, by leveraging the characteristics observed for both benign and adversarial data, we apply binary classifiers, including simple threshold-based classification, ensembles of these simple classifiers, and neural networks. In an extensive analysis of different state-of-the-art ASR systems and language data sets, we demonstrate the supreme performance of this approach, receiving a mean area under the receiving operator characteristic (AUROC) for distinguishing adversarial examples against clean and noisy data higher than 99% and 98%, respectively. To assess the robustness of our method, we propose adaptive attacks that are constructed with an awareness of the defense mechanism in place. This results in a decrease in the AUROC, but at the same time, the adversarial clips become noisier, which makes them easier to detect through filtering and creates another avenue for preserving the system's robustness.

## 1 INTRODUCTION

Voice recognition technologies are widely used in the devices that we interact with daily—in smartphones or virtual assistants—and are also being adapted for more safety-critical tasks like self-driving cars (Wu et al., 2022) and healthcare applications. Safeguarding these systems from malicious attacks thus plays a more and more critical role, e.g., manipulated erroneous transcriptions can potentially lead to breaches in customer security. Another example involves the targeting of commercial speech recognition devices like Google Assistant, Google Home, Microsoft Cortana, and Amazon Echo using over-the-air attacks. They use substitute models to mimic the unknown target model, aiming to make the system recognize their desired inputs (Chen et al., 2020). By modifying an audio signal for the Kaldi ASR system, for example, the system could output a false transcription containing the command to purchase a product (Schönherr et al., 2019).

State-of-the-art ASR systems are based on deep learning (Kahn et al., 2020; Chung et al., 2021). Unfortunately, deep neural networks (NN) are highly vulnerable to adversarial attacks, since the inherent properties of the model make it easy to generate an input that is necessarily mislabeled, simply by incorporating a low-level additive perturbation (Szegedy et al., 2014; Goodfellow et al., 2015; Ilyas et al., 2019; Du et al., 2020). A well-established method to generate adversarial examples (AE), which is also applicable to ASR systems, is the Carlini & Wagner (C&W) attack (Carlini & Wagner, 2018). It aims to minimize a perturbation $\delta$ that—when added to a benign audio signal $x$—induces the system to recognize a phrase chosen by the attacker. The psychoacoustic attack (Schönherr et al., 2019; Qin et al., 2019) specifically developed for ASR systems goes one step further than the C&W attack. By considering principles of acoustic perception, it creates an

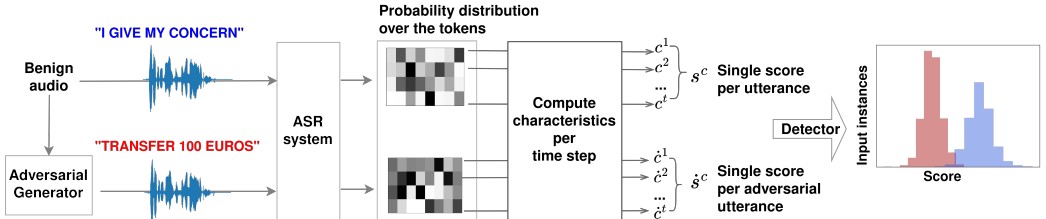

Figure 1: Proposed workflow to identify AEs: (1) compute output probability distribution characteristics per time step, (2) use a detector to tell benign and AEs apart. $c^1$ to $c^T$ and $\dot{c}^1$ to $\dot{c}^T$ represent benign and adversarial input characteristics, while $s^c$ and $\dot{s}^c$ denote final scores.

inconspicuous disturbance $\delta$ utilizing time-frequency masking, i.e., it shapes the perturbations to fall below the estimated time-frequency masking threshold of human listeners, rendering $\delta$ hardly perceptible, and sometimes even *inaudible* to humans.

Motivated by the security gap of ASR in the presence of adversarial attacks, in this work, we introduce a novel detection technique to distinguish benign from adversarial data by analyzing the distribution of tokens generated by an ASR system at each output step. Our method relies on the observed statistical characteristics of attacked samples and trains Gaussian classifiers (GC), ensemble models, and NN using these as features. To assess the generality of our findings, we evaluate our method's performance across diverse state-of-the-art ASR models and datasets that cover a range of languages. Empirical results confirm that the proposed detection technique effectively differentiates between benign and targeted adversarial data, achieving an AUROC exceeding 99% in all tested end-to-end (E2E) models. To assess the effectiveness of our defense in more challenging scenarios, we test our classifiers when faced with noisy audio data, untargeted attacks, and create adaptive adversarial samples, assuming the attacker to have complete knowledge about the defense mechanism. While the classifiers demonstrate robustness w.r.t. noise, they are vulnerable to adaptive attacks. However, as the resulting adversarial audio files are more distorted, they are easier to spot for human ears and identifiable using filtering techniques. We demonstrate that our approach surpasses the leading temporal dependency technique and the noise flooding method by achieving an improvement in all test data. Moreover, our method is suitable for use with any ASR system that forecasts a probability distribution over output tokens at each time step, and it eliminates the need for supplementary data preprocessing, adversarial training augmentation, or model fine-tuning.

## 2 RELATED WORK

When it comes to mitigating the impact of adversarial attacks, there are two main research directions. On the one hand, there is a strand of research dedicated to enhancing the robustness of models. On the other hand, there is a separate research direction that focuses on designing detection mechanisms to recognize the presence of adversarial attacks.

Concerning the robustness of models, there are diverse strategies, one of which involves modifying the input data within the ASR system. This concept has been adapted from the visual to the auditory domain. Examples of input data modifications include quantization, temporal smoothing, down-sampling, low-pass filtering, slow feature analysis, and auto-encoder reformation (Meng & Chen, 2017; Guo et al., 2018; Pizarro et al., 2021). However, these techniques become less effective once integrated into the attacker's deep learning framework (Yang et al., 2019). Another strategy to mitigate adversarial attacks is to accept their existence and force them to be *perceivable* by humans (Eisenhofer et al., 2021), with the drawback that the AEs can continue misleading the system. Adversarial training (Madry et al., 2018), in contrast, involves employing AEs during training to enhance the NN's resiliency against adversarial attacks. Due to the impracticality of covering all potential attack classes through training, adversarial training has major limitations when applied to large and complex data sets, such as those commonly used in speech research (Zhang et al., 2019). Additionally, this approach demands high computational costs and can result in reducing the accuracy on benign data. A recent method borrowed from the field of image recognition is adversarial purification, where generative models are employed to cleanse the input data prior to inference

(Yoon et al., 2021; Nie et al., 2022). However, only a few studies have investigated this strategy within the realm of audio. Presently, its ASR applications are confined to smaller vocabularies, and it necessitates substantial computational resources, while also resulting in decreased accuracy when applied to benign data (Wu et al., 2023).

In the context of improving the discriminative power against adversarial attacks, Rajaratnam & Kalita (2018) introduced a noise flooding (NF) method that quantifies the random noise needed to change the model's prediction, with smaller levels observed for AEs. However, NF was only tested against a specific untargeted attack on a 10-word speech classification system. A prominent non-differentiable approach uses the inherent temporal dependency (TD) in raw audio signals (Yang et al., 2019). This strategy requires a minimal length of the audio stream for optimal performance. Unfortunately, Zhang et al. (2020) successfully evaded the detection mechanism of TD by preserving the necessary temporal correlations, leading to the generation of robust AEs once again. Däubener et al. (2020) proposed AEs detection for hybrid ASR systems based on uncertainty measures. They applied their method to a limited vocabulary tailored for digit recognition. Two of these uncertainty metrics—the mean Kullback-Leibler divergence (KLD) and mean entropy—are also among those characteristics of the output distribution that we investigate, next to many others, for constructing defenses against AEs in this paper. It's worth noting that Meyer et al. (2016) also utilized the averaged KLD between the output distributions of consecutive time-steps (which they referred to as mean temporal distance), but to assess the reliability of an ASR output over time.

## 3 BACKGROUND

**Adversarial attacks**  In order to keep things convenient, we assume that the label transcript $y$ and the input audio signal $x$ are related by $y = f(x)$, where $f(\cdot)$ refers to the ASR system's function, which maps an audio input to the sequence of words it most likely contains. To create a targeted AE, we need to find a small perturbation $\delta$ of the input that causes the ASR system to predict the desired transcript $\hat{y}$ given $x + \delta$, i.e., $f(x + \delta) = \hat{y} \neq y = f(x)$. This perturbation $\delta$ is usually constructed by gradient descent-based minimization of the following function

$$l(x, \delta, \hat{y}) = l_t(f(x + \delta), \hat{y}) + c \cdot l_a(x, \delta) \ , \tag{1}$$

which includes two loss functions: (1) a task-specific loss, $l_t(\cdot)$, to find a distortion that induces the model to output the desired transcription target $\hat{y}$, and (2) an acoustic loss, $l_a(\cdot)$, that is used to make the noise $\delta$ smaller in energy and/or imperceptible to human listeners. In the initial steps of the iterative optimization procedure, the weighting parameter $c$ is usually set to small values to first find a viable AE. Later, $c$ is often increased, in order to minimize the distortion, to render it as inconspicuous as possible.

The most common targeted attacks for audio are the C&W Attack and Qin's *Imperceptible Attack*, two well-established optimization-based adversarial algorithms. These techniques have proven successful in targeted attacks and offer a publicly available PyTorch implementation. In the C&W attack (Carlini & Wagner, 2018), $l_t$ is the negative log-likelihood of the target phrase and $l_a = |\delta|_2^2$. Moreover, $|\delta|$ is constrained to be smaller than a predefined value $\epsilon$, which is decreased step-wise in an iterative process. The *Imperceptible Attack* (Qin et al., 2019) is divided into two stages. The first stage of the attack follows the approach outlined by C&W. The second stage of the algorithm aims to decrease the perceptibility of the noise by using frequency masking, following psychoacoustic principles. Moreover, several untargeted attacks have been proposed. These include the projected gradient descent (PGD) (Madry et al., 2018), a well-known optimization-constrained method, as well as two model-independent attacks—the Kenansville attack (Abdullah et al., 2020; 2021) utilizing signal processing methods, and the genetic attack (Alzantot et al., 2018), a gradient-free optimization algorithm.

**End-to-end ASR systems**  An E2E ASR system (Prabhavalkar et al., 2023) can be described as a unified ASR model that directly transcribes a speech waveform into text, as opposed to orchestrating a pipeline of separate ASR components. Here, the system directly converts a sequence of acoustic input features into a sequence of tokens (e.g., phonemes, characters, or words). Ideally, E2E ASR models are fully differentiable and thus can be trained end-to-end by maximizing the conditional log-likelihood with respect to the desired output. Various E2E ASR models follow an encoder-only or an encoder-decoder architecture and typically are built using recurrent neural network (RNN) or

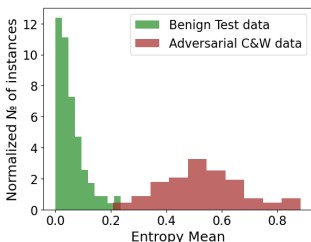 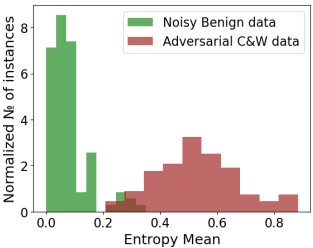 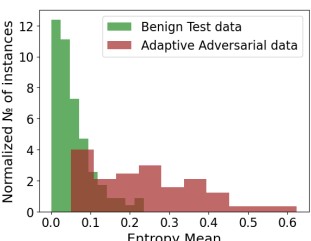

(a) Benign test-clean data vs. C&W AEs.

(b) Noisy benign data vs. C&W AEs.

(c) Benign test-clean data vs. adaptive C&W AEs.

Figure 2: Mean entropy histograms of 100 data samples vs. 100 C&W AEs.

transformer layers. Special care is taken of the unknown temporal alignments between the input waveform and output text, where the alignment can be modeled explicitly (e.g., CTC (Graves et al., 2006), RNN-T (Graves, 2012)), or implicitly using attention (Watanabe et al., 2017). Furthermore, language models can be integrated in order to improve prediction accuracy by considering the most probable sequences (Toshniwal et al., 2018).

## 4    OUTPUT DISTRIBUTION-BASED DEFENSE APPROACH

We propose to leverage the probability distribution over the tokens from the output vocabulary in order to identify adversarial attacks. A schematic of our approach is displayed in Fig. 1. An audio clip–either benign or malicious–is fed to the ASR system. The system then generates probability distributions over the output tokens in each time step. The third step is to compute pertinent characteristics of these output distributions, as detailed below. Then, we use a function (i.e., the mean, median, maximum, or minimum) to aggregate the values of the characteristics to a single score per utterance. Lastly, we employ a binary classifier for differentiating adversarial instances from test data samples.

**Characteristics of the output distribution**    For each time step $t$, the ASR system produces a probability distribution $p^{(t)}$ over the tokens $i \in \mathcal{V}$ of the output vocabulary $\mathcal{V}$. For an output utterance of length $T$ we compute the following quantities of this distribution for every $t \in \{1, \ldots, T\}$: the **median** of $p^{(t)}(i), i = 1, 2, \ldots, |V|$, the **minimum** $\min_{i \in \{1,\ldots,|V|\}} p^{(t)}(i)$, the **maximum** $\max_{i \in \{1,\ldots,|V|\}} p^{(t)}(i)$, the **Shannon entropy** $H(p^{(t)}) = -\sum_{i=1}^{|\mathcal{V}|} p^{(t)}(i) \cdot \log p^{(t)}(i)$, the **Kullback–Leibler divergence (KLD)** between the output distributions in two successive time steps

$$D_{\mathrm{KL}}(p^{(t)} \| p^{(t+1)}) = \sum_{i=1}^{|\mathcal{V}|} p^{(t)}(i) \cdot \log \frac{p^{(t)}(i)}{p^{(t+1)}(i)},$$ and the **Jensen-Shannon divergence (JSD)** between the output distributions in two successive time steps, which is obtained as a symmetrized alternative of the Kullback-Leibler divergence

$$D_{\mathrm{JSD}}(p^{(t)}, p^{(t+1)}) = \frac{1}{2} D_{\mathrm{KL}}(p^{(t)} \| M) + \frac{1}{2} D_{\mathrm{KL}}(p^{(t+1)} \| M), \text{ where } M = \frac{1}{2}(p^{(t)} + p^{(t+1)}) \ .$$

We aggregated the step-wise median, minimum, maximum, and entropy over all steps $t = 1, \ldots T$ of the output sequence into a single score by taking the mean, median, minimum, or maximum w.r.t. the respective values for different time steps $t$.

**Binary classifier**    The extracted characteristics of the output distribution can then be used as features for a binary classifier. An option to obtain simple classifiers is to fit a Gaussian distribution to each score computed for the utterances from a held-out set of benign data. If the probability of a new audio sample is below a chosen threshold, this example is classified as adversarial. For illustration, Fig. 2 displays histograms of the mean entropy values for the LSTM-LAS-CTC model's predictive distribution over benign and adversarial data using LibriSpeech. A more sophisticated approach is to employ ensemble models (EM), in which multiple Gaussian distributions, fitted to a single score each, produce a unified decision by a majority vote. Another option is to construct an NN that takes all the characteristics described above as input.

**Adaptive attack**   An adversary with complete knowledge of the defense strategy can implement so-called adaptive attacks. In order to show the advantage of our proposed defense, we analyze the options for adaptive AEs. For this, we construct a new loss $l_k$ by adding a penalty $l_s$ to the loss function in equation 1, weighted with some factor $\alpha$:

$$l_k(x, \delta, \hat{y}) = (1 - \alpha) \cdot l(x, \delta, \hat{y}) + \alpha \cdot l_s^c(x) \ . \tag{2}$$

When attacking a Gaussian classifier that is based on characteristic $c$, $l_s$ corresponds to the $L_1$ norm of the difference between the mean $\overline{s}^c$ of the Gaussian fitted to the respective scores of benign data (resulting from aggregating $c$ over each utterance) and the score of $x$. When attacking an EM, $l_s$ is set to

$$l_s(x) = \sum_{i=1}^{T} |\overline{s}^{c_i} - s^{c_i}(x)| \ ,$$

where $c_1 \ldots c_T$ corresponds to the characteristics used by the Gaussian classifiers of the ensemble is composed of. In the case of NNs, $l_s(x)$ is simply the $L_1$ norm, quantifying the difference between the NN's predicted outcome (a probability value) and one (indicating the highest probability for the benign category).

## 5   EXPERIMENTS

### 5.1   EXPERIMENTAL SETTINGS

**Datasets**   We use the LibriSpeech dataset (Panayotov et al., 2015) comprises approximately 1,000 hours of English speech, sampled at a rate of 16KHz, extracted from audiobooks. We further use Aishell (Bu et al., 2017), an open-source speech corpus for Mandarin Chinese. Since Chinese is a tonal language, the speech of this corpus exhibits significant and meaningful variations in pitch. Additionally, we consider the Common Voice (CV) corpus (Ardila et al., 2020), one of the largest multilingual open-source audio collections available in the public domain. Created through crowd-sourcing, CV includes additional complexities within the recordings, such as background noise and reverberation.

**ASR systems**   We analyzed fully integrated Pytorch-based deep learning end-to-end speech engines. In order to assess the versatility of our method, which relies on identifying specific characteristics in the system response to attacked samples, we trained various ASR models on different datasets and languages such as English, German, Italian, and Mandarin. These models generate diverse output formats, depending on their tokenizer selection, which can encode either characters or subwords. Specifically, the models we use produce output structures with neuron counts of 32, 500, 1,000, 5,000, or 21,128.

We investigate three different models. The first employs a wav2vec2 encoder (Baevski et al., 2020) and a CTC decoder. The second integrates an encoder, a decoder, and an attention mechanism between them, as initially proposed with the Listen, Attend, and Spell (LAS) system (Chan et al., 2016), employing a CRDNN encoder and a LSTMs decoding (Chorowski et al., 2015). The third model implements a transformer architecture relying on attention mechanisms for both encoding and decoding (Vaswani et al., 2017; Wolf et al., 2020). The models are shortly referred to as wav2vec, LSTM, and Trf, respectively, in our tables. To improve generalization, we applied standard data augmentation techniques provided in SpeechBrain: corruption with random samples from a noise collection, removing portions of the audio, dropping frequency bands, and resampling the audio signal at a slightly different rate.

**Adversarial attacks**   To generate the AEs, we utilized a repository that contains a PyTorch implementation of all considered attacks (Olivier & Raj, 2022). We randomly selected 200 samples from the test set, with 100 of them designated for testing purposes. For targeted attacks, each of these samples was assigned a new adversarial target transcript sourced from the same dataset. Our selection process adhered to four guiding principles: (1) the audio file's original transcription cannot be used as the new target description, (2) there should be an equal number of tokens in both the original and target transcriptions, (3) each audio file should receive a unique target transcription, and (4) audio clips must be no longer than five seconds. We reduced the audio clip

Table 1: Comparison of the performance of ASR systems on benign and noisy data, in terms of word and sentence error rate on 100 utterances. LM denotes the language model.

| | | | Benign data | | Noisy data | | | |
|---|---|---|---|---|---|---|---|---|
| Model | Language | LM | WER | SER | WER | SER | SNR$_{Seg}$ | SNR |
| LSTM | Italian (It) | ✗ | 15.65% | 52% | 31.74% | 72% | -3.65 | 6.52 |
| LSTM | English (En) | ✗ | 5.37% | 31% | 8.46% | 45% | 2.75 | 6.67 |
| LSTM | English (En-LM) | ✓ | 4.23% | 24% | 5.68% | 30% | 2.75 | 6.67 |
| wav2vec | Mandarin (Ma) | ✓ | 4.37% | 28% | 8.49% | 43% | 5.25 | 4.50 |
| wav2vec | German (Ge) | ✗ | 8.65% | 33% | 16.08% | 51% | -2.66 | 7.85 |
| Trf | Mandarin (Ma) | ✗ | 4.79% | 29% | 7.40% | 40% | 5.25 | 4.50 |
| Trf | English (En) | ✓ | 3.10% | 20% | 11.87% | 44% | 2.75 | 6.67 |

Table 2: Quality of 100 generated C&W, Psychoacoustic, and adaptive attacks, measured by the average performance of the ASR systems across all models w.r.t the target utterances as well as the SNRs. Adaptive attack customized to target a GC optimized for the most effective characteristic.

| | C&W attack | | | | Psychoacoustic attack | | | | Adaptive attack | | | |
|---|---|---|---|---|---|---|---|---|---|---|---|---|
| Model | WER | SER | SNR$_{Seg}$ | SNR | WER | SER | SNR$_{Seg}$ | SNR | WER | SER | SNR$_{Seg}$ | SNR |
| LSTM (It) | 0.84% | 3.00% | 17.79 | 44.51 | 0.84% | 3.00% | 18.17 | 38.52 | 0.84% | 3.00% | -1.47 | 18.36 |
| LSTM (En) | 1.09% | 2.00% | 14.91 | 33.29 | 1.09% | 2.00% | 15.14 | 31.92 | 0.30% | 1.00% | 0.23 | 14.01 |
| LSTM (En-LM) | 1.19% | 2.00% | 17.50 | 36.46 | 1.19% | 2.00% | 17.82 | 33.93 | 0.40% | 1.00% | 3.18 | 16.82 |
| wav2vec (Ma) | 0.08% | 1.00% | 22.22 | 31.35 | 0.08% | 1.00% | 22.73 | 30.66 | 0.08% | 1.00% | -4.30 | 4.09 |
| wav2vec (Ge) | 0.00% | 0.00% | 20.58 | 50.86 | 0.00% | 0.00% | 21.08 | 41.46 | 0.00% | 0.00% | -12.96 | 10.88 |
| Trf (Ma) | 0.00% | 0.00% | 31.93 | 49.35 | 0.00% | 0.00% | 29.47 | 32.69 | 0.00% | 0.00% | -1.09 | 8.01 |
| Trf (En) | 0.00% | 0.00% | 27.85 | 53.54 | 0.00% | 0.00% | 25.70 | 37.68 | 0.00% | 0.00% | -0.19 | 14.69 |

length to save time and resources, as generating AEs for longer clips can take up to an hour, depending on the computer and model complexity (Carlini & Wagner, 2018). A 5-sec length was a favorable trade-off between time/resources, and the number of AEs created per model. A selection of benign, adversarial, and noisy data employed in our experiments are available online at `https://confunknown.github.io/characteristics_demo_AEs/`.

We initialize the adaptive attack with inputs that are already misleading the system. Then, to generate adaptive AEs, we follow the approach of minimizing the loss function described in equation 2 and execute 1,000 additional iterations on 100 randomly chosen AEs. We evaluate the adaptive attacks by keeping the $\alpha$ value constant at a value of 0.3, while the $\delta$ factor, which is gradually reduced in an iterative manner to reduce noise, remains unchanged during the initial 500 iterations. This approach noticeably diminishes the discriminative capability of our defense across all models. However, this reduction in discriminative power comes at the expense of generating noisy data that, as evidenced by our experimental results in Subsection 5.3, can be easily detected through filtering. Further experiments were carried out using different configurations; these changes resulted in data with lower noise levels but also led to weaker attacks. Detailed outcomes of these experiments are available in the App. A.1.

**Adversarial example detectors** We construct three kinds of binary classifiers: Based on the 24 single scores, we obtain 24 simple Gaussian classifiers (GC) per model. To construct an ensemble model, we implement a majority voting technique, utilizing a total of $T \in \{3, 5, 7, 9\}$ GCs. The choice of which GCs to incorporate is determined by evaluating the performance of each characteristic across all models and ranking them in descending order based on the results of the validation set. The outcome of the ranking can be found in App. A.2. The neural network architecture consists of three fully connected layers, each with 72 hidden nodes, followed by an output layer. We employ a sigmoid activation function to generate a probability output in the range of 0 to 1 that can be converted to class values. The network is trained using ADAM optimization (Kingma & Ba, 2015) with a learning rate of 0.0001 for 250 epochs. Running the assessment with our detectors took approx. an extra 18.74 ms per sample, utilizing an NVIDIA A40 with a memory capacity of 48 GB, see App. A.3 for more details.

## 5.2 QUALITY OF ASR SYSTEMS AND ADVERSARIAL ATTACKS

To assess the quality of the trained models as well as the performance of the AEs, we measured the word error rate (WER), the character error rate (CER), the sentence error rate (SER), the Signal-

Table 3: Quality of 100 generated PGD, Genetic, and Kenansville attacks, measured by the average performance of the ASR systems across all models w.r.t the true labels as well as the SNRs.

| Model | PGD attack | | | | Genetic attack | | | | Kenansville attack | | | |
|---|---|---|---|---|---|---|---|---|---|---|---|---|
| | WER | SER | $SNR_{Seg}$ | SNR | WER | SER | $SNR_{Seg}$ | SNR | WER | SER | $SNR_{Seg}$ | SNR |
| LSTM (It) | 121% | 100% | 7.39 | 25.76 | 41.6% | 83.0% | 3.04 | 35.13 | 73.2% | 95.0% | -6.1 | 6.32 |
| LSTM (En) | 95% | 100% | 15.13 | 25.91 | 24.5% | 85.0% | 6.49 | 33.59 | 49.8% | 85.0% | 1.32 | 7.4 |
| LSTM (En-LM) | 100% | 100% | 15.19 | 26.21 | 23.8% | 83.0% | 6.63 | 33.59 | 49.3% | 78.0% | 1.32 | 7.4 |
| wav2vec (Ma) | 90% | 100% | 20.09 | 23.68 | 36.2% | 94.0% | 6.24 | 23.74 | 99.0% | 99.0% | 6.18 | 6.33 |
| wav2vec (Ge) | 102% | 100% | 6.88 | 26.79 | 30.7% | 78.0% | 1.72 | 33.39 | 49.3% | 86.0% | -5.73 | 6.89 |
| Trf (Ma) | 126% | 100% | 19.49 | 26.41 | 44.1% | 96.0% | 4.36 | 23.74 | 73.8% | 98.0% | 6.18 | 6.33 |
| Trf (En) | 102% | 100% | 14.88 | 26.58 | 17.8% | 77.0% | 8.79 | 33.59 | 40.7% | 72.0% | 1.32 | 7.4 |

to-Noise Ratio (SNR), and the Segmental Signal-to-Noise Ratio ($SNR_{Seg}$). The latter measures the adversarial noise energy in Decibels and considers the entire audio signal. Thus, a higher $SNR_{Seg}$ indicates less additional noise. Specific information about each of these formulas is available in the App. A.4.

**Quality of ASR systems** Tab. 16 in the App. reports the results achieved with different Speech-Brain recipes on all datasets. The performance is consistent with those documented by Ravanelli et al. (2021), where you can also find detailed hyperparameter information for all these models. To determine the classifier's effectiveness in a situation that better mimics reality, 100 benign audio clips are contaminated with background noise. This involves introducing random samples from a noise dataset into the speech signal. The noise instances are randomly sampled from the Freesound section of the MUSAN corpus (Snyder et al., 2015; Ko et al., 2017), which includes room impulse responses, as well as 929 background noise recordings. We utilize SpeechBrain's environmental corruption function to add noise to the input signal. Tab. 1 presents the performance of the ASR systems for noisy data, utilizing a total of 100 utterances. The impact on system performance is evident, resulting in a significant rise in WER due to the low SNR ratio.

**Quality of adversarial attacks** To estimate the effectiveness of the targeted adversarial attacks we measured the error w.r.t. the target utterances, reported in Tab. 2. We achieved nearly 100% success in generating targeted adversarial data for all attack types across all models. The model with the lowest average SNR distortion registered at 31.35 dB, while the highest—i.e., the least distorted—was 53.54 dB. In a related study by Carlini & Wagner (2018), they reported a mean distortion of 31 dB. In contrast, for untargeted attacks, we measured the error relative to the true label, the higher the WER the stronger the attack. We consider it a genuine threat as one where the attack produces a WER of at least 50%, surpassing the impact influence of background noise. Diverse settings were explored in our experiments, and these are detailed in the App. A.6. Both PGD and Kenansville regulate the distortion of the attack using an SNR factor to limit the perturbations, but in different ways, with PGD achieving optimal results at a factor of 25, while Kenansville performed best at a factor of 10. In the case of a genetic attack, we found a minimal effect on the WER, failing to reach 50% across all models, these results are presented in Tab. 3. In general, our findings are in line with the results discussed by Olivier & Raj (2022). When generating AEs with the proposed adaptive adversarial attack, we also managed to achieve an almost 100% success rate, see Tab. 2. However, the AEs turned out to be much noisier, as displayed by a maximum average SNR value of 18.36 dB when comparing all models. This makes the perturbations more easily perceptible to humans.

## 5.3 PERFORMANCE OF ADVERSARIAL EXAMPLE DETECTORS

**Detecting C&W and Psychoacoustic attacks** To distinguish benign audio clips from malicious inputs, we calculate the characteristic scores and use them to train binary classifiers as described in Sec. 4. The detection performance of our classifiers w.r.t C&W and Psychoacoustic attacks are quite similar. Therefore, we present the C&W results in Tab. 4 and include the Psychoacoustic results in the App. A.7. We contrast our binary classifiers with NF and TD. For the GC, we report for each model the performance for the characteristic best-performing on the validation set (detailed results for all other characteristics can be found in the App. A.8). Our findings show that the proposed binary classifiers consistently outperform NF and TD across all models when distinguishing between benign and adversarial data, achieving an impressive discrimination accuracy of over 99% in every case, regardless of the deep learning architecture used by the ASR system, the data it was trained on,

Table 4: Comparing classifiers on clean and noisy data, evaluating AUROC for all models using 100 samples from the clean test set and 100 C&W AEs. $^{(*)}$ denotes best-performing score-characteristic.

| Model | Score-Characteristic[(*)] | Noisy vs. C&W adversarial data | | | | Benign vs. C&W adversarial data | | | |
|---|---|---|---|---|---|---|---|---|---|
| | | NF | TD | GC | NN | NF | TD | GC | NN |
| LSTM (It) | Mean-Median | 0.9218 | 0.8377 | **0.9686** | 0.9557 | 0.8762 | 0.8923 | **0.9980** | 0.9962 |
| LSTM (En) | Mean-Median | 0.9289 | 0.9695 | 0.9966 | **0.9996** | 0.8868 | 0.9697 | **0.9993** | 0.9992 |
| LSTM (En-LM) | Max-Max | 0.9680 | 0.9022 | 0.9835 | **0.9875** | 0.9345 | 0.9293 | 0.9828 | **0.9903** |
| wav2vec (Ma) | Mean-Entropy | 0.9406 | **0.9817** | 0.9578 | 0.9680 | 0.8993 | 0.9937 | **0.9947** | 0.9937 |
| wav2vec (Ge) | Max-Min | 0.9372 | 0.9557 | 0.9652 | **0.9992** | 0.8725 | 0.9836 | **0.9941** | 0.9910 |
| Trf (Ma) | Median-Max | 0.9572 | 0.9790 | **0.9864** | 0.9803 | 0.9243 | 0.9828 | **0.9978** | 0.9969 |
| Trf (En) | Max-Median | 0.9702 | 0.9448 | 0.9287 | **0.9844** | 0.8998 | 0.9828 | **1.0000** | 1.0000 |
| Average AUROC across all models | | 0.9462 | 0.9386 | 0.9695 | **0.9821** | 0.8990 | 0.9620 | 0.9952 | **0.9953** |

Table 5: Classification accuracies for classifiers, based on a threshold for a maximum 1% FPR (if possible) and a minimum 50% TPR, using 100 benign data and 100 C&W AEs.

| Model | TD | GC | EM=3 | EM=5 | EM=7 | EM=9 | NN |
|---|---|---|---|---|---|---|---|
| LSTM (It) | 72.50% / 0.05 | **98.00% / 0.01** | 94.00% / 0.01 | 92.50% / 0.01 | 90.50% / 0.01 | 91.50% / 0.01 | 95.00% / 0.01 |
| LSTM (En) | 85.00% / 0.01 | 98.50% / 0.01 | **99.50% / 0.00** | **99.50% / 0.00** | **99.50% / 0.00** | **99.50% / 0.00** | 98.50% / 0.00 |
| LSTM (En-LM) | 74.00% / 0.02 | 90.50% / 0.01 | 91.00% / 0.01 | 92.50% / 0.01 | **97.00% / 0.01** | 92.50% / 0.01 | 95.00% / 0.01 |
| wav2vec (Ma) | 97.00% / 0.01 | 97.50% / 0.01 | 97.50% / 0.01 | 96.00% / 0.00 | 91.50% / 0.00 | 93.50% / 0.00 | **98.50% / 0.01** |
| wav2vec (Ge) | 94.00% / 0.03 | **98.00% / 0.01** | 97.00% / 0.01 | **98.00% / 0.00** | 96.50% / 0.00 | **98.00% / 0.00** | 97.00% / 0.01 |
| Trf (Ma) | 96.50% / 0.01 | **98.00% / 0.01** | 96.00% / 0.01 | 96.50% / 0.00 | 96.00% / 0.00 | 96.00% / 0.01 | 95.50% / 0.01 |
| Trf (En) | 93.00% / 0.01 | 99.50% / 0.01 | **100.0% / 0.00** | **100.0% / 0.00** | **100.0% / 0.00** | **100.0% / 0.00** | **100.0% / 0.00** |
| Avg. accuracy / FPR | 87.43% / 0.02 | **97.14% / 0.01** | 96.43% / 0.01 | 96.43% / 0.00 | 95.86% / 0.00 | 95.86% / 0.00 | 97.07% / 0.01 |

and whether it employs a language model during decoding or not. In a more challenging context, where distinguishing between noisy and adversarial data, our proposed defense still surpasses NF and TD for all models except for one. We observe that among all classifiers, the NN stands out as the most robust when comparing noisy and benign data scenarios, showing only a minimal decrease of 1.42% in the average AUROC across all models. It's worth noting that the performance of TD on noisy data hasn't been analyzed before, and former investigations were limited to the English language (Yang et al., 2019). Similarly, NF was solely tested against the untargeted genetic attack in a 10-word classification system. Some characteristics perform consistently well, independently of the adversarial data, and only benign data is needed for choosing the threshold. This is displayed by the results for GCs based on the mean-median for both targeted attacks in the first two columns of Tab. 6. Moreover, even the neural network solely trained on C&W attacks performs equally well against Psychoacoustic AEs. These results indicate a good transferability to other kinds of targeted attacks.

To evaluate the goodness-of-fit performance of our classifiers, we adopted a conservative threshold selection criterion: the highest false positive rate (FPR) below 1% (if available) while maintaining a minimum true positive rate (TPR) of 50%. This evaluation considers EMs with different total voting values $T \in \{3, 5, 7, 9\}$. Hence, our classifiers consistently achieve a high average accuracy exceeding 95%, surpassing the performance of TD, as indicated in Tab. 5. We suggest opting for an EM approach, which tends to minimize variance, or an NN that apart from minimizing variance has the potential for enhanced generalization with further refinements. Additional goodness-of-fit measurements across all models are available in the App. A.9.

**Detecting untargeted attacks** To assess the transferability of our detectors to untargeted attacks, we investigated the defense performance of GCs based on the mean-median characteristic and NNs trained on C&W AEs when exposed to PGD, Geneticc, or Kenansville attacks. Results are reported in Tab. 6. While the detection performance decreases in comparison to targeted attacks, our methods are still way more efficient than TD, with AUROCs even exceeding 90% for the Kenansville attack. In general, the Genetic attack proves challenging to detect, which may be attributed to its limited impact on the WER (compare Tab. 3). Advantageously, limited research addresses untargeted attacks in large-vocabulary ASR systems, in general, they are less threatening and all instances we investigated are characterized by noise, making them easily noticeable by human hearing.

Table 6: AUROC assessment to detect AEs using GCs and NNs across various attacks.

| Model | C&W TD | GC | NN | Psychoacoustic TD | GC | NN | PGD TD | GC | NN | Genetic TD | GC | NN | Kenansville TD | GC | NN |
|---|---|---|---|---|---|---|---|---|---|---|---|---|---|---|---|
| LSTM (It) | 0.89 | 1.00 | 1.00 | 0.89 | 1.00 | 1.00 | 0.71 | 0.94 | 0.94 | 0.54 | 0.58 | 0.68 | 0.68 | 0.82 | 0.92 |
| LSTM (En) | 0.97 | 1.00 | 1.00 | 0.97 | 1.00 | 1.00 | 0.69 | 0.96 | 0.98 | 0.53 | 0.50 | 0.68 | 0.76 | 0.89 | 0.89 |
| LSTM (En-LM) | 0.93 | 0.95 | 0.99 | 0.94 | 0.96 | 0.99 | 0.83 | 1.00 | 0.78 | 0.55 | 0.64 | 0.69 | 0.76 | 0.79 | 0.89 |
| wav2vec (Ma) | 0.99 | 0.99 | 0.99 | 0.99 | 0.99 | 0.99 | 0.77 | 0.84 | 0.84 | 0.65 | 0.67 | 0.73 | 0.89 | 0.97 | 0.97 |
| wav2vec (Ge) | 0.98 | 1.00 | 0.99 | 0.98 | 1.00 | 0.99 | 0.82 | 0.78 | 0.40 | 0.55 | 0.65 | 0.58 | 0.77 | 0.93 | 0.85 |
| Trf (Ma) | 0.98 | 0.99 | 1.00 | 0.99 | 0.99 | 0.99 | 0.86 | 0.76 | 0.91 | 0.59 | 0.65 | 0.84 | 0.90 | 1.00 | 1.00 |
| Trf (En) | 0.98 | 1.00 | 1.00 | 0.99 | 1.00 | 1.00 | 0.75 | 0.75 | 0.59 | 0.53 | 0.46 | 0.64 | 0.73 | 0.89 | 0.94 |
| Avg. | 0.96 | 0.99 | **1.00** | 0.97 | **0.99** | **0.99** | 0.78 | **0.86** | 0.78 | 0.56 | 0.59 | **0.69** | 0.79 | 0.90 | **0.92** |

Table 7: Evaluating filtering to preserve system robustness in accuracy with 100 clean test set samples and 100 adaptive C&W AEs, using a threshold aiming for a maximum 1% FPR when feasible.

| Model | Adaptive AE attack performance pre-filtering GC | EM=9 | NN | Filtering AEs aiming a GC LPF: WER / CER | SG: WER / CER | Filtering AEs aiming an EM=9 LPF: WER / CER | SG: WER / CER | Filtering AEs aiming a NN LPF: WER / CER | SG: WER / CER |
|---|---|---|---|---|---|---|---|---|---|
| LSTM (It) | 33.50 | 50.50 | 53.50 | 76.00 / 78.50 | 82.00 / 94.00 | 74.00 / 79.00 | 74.50 / 88.00 | 79.50 / 86.00 | 88.50 / 97.50 |
| LSTM (En) | 29.50 | 50.50 | 73.50 | 86.00 / 88.00 | 98.00 / 100.0 | 98.00 / 97.50 | 96.00 / 99.00 | 86.50 / 89.00 | 99.00 / 100.0 |
| LSTM (En-LM) | 42.50 | 44.50 | 67.50 | 80.50 / 86.50 | 99.00 / 99.50 | 76.00 / 80.50 | 94.50 / 95.00 | 81.00 / 85.50 | 99.50 / 99.50 |
| wav2vec (Ma) | 37.50 | 58.00 | 49.50 | 99.50 / 99.50 | 96.00 / 96.00 | 100.0 / 100.0 | 96.00 / 96.00 | 100.0 / 100.0 | 99.50 / 99.50 |
| wav2vec (Ge) | 25.50 | 71.00 | 60.50 | 96.00 / 98.00 | 97.50 / 96.00 | 96.50 / 98.50 | 98.00 / 98.00 | 97.50 / 98.00 | 98.00 / 97.50 |
| Trf (Ma) | 28.50 | 40.00 | 26.00 | 74.00 / 74.00 | 79.50 / 79.50 | 71.50 / 71.50 | 74.50 / 74.50 | 82.00 / 82.00 | 89.50 / 89.50 |
| Trf (En) | 25.00 | 25.00 | 25.50 | 75.00 / 75.00 | 91.00 / 93.50 | 50.00 / 50.00 | 84.00 / 91.00 | 96.50 / 97.50 | 100.0 / 100.0 |
| Avg. accuracy | 31.71 | 48.50 | 50.86 | 83.86 / 85.64 | 91.86 / **94.07** | 80.86 / 82.43 | 88.21 / **91.64** | 89.00 / 91.14 | 96.29 / **97.64** |

**Detecting adaptive adversarial attacks** The accuracy of our classifiers experiences a substantial decline across all models due to adaptive attacks when evaluated with a threshold aiming for a maximum FPR of 1% (where feasible). That means, that the defense provided gets ineffective if its usage is known to the attacker. However, one can leverage the fact, that the adaptive attack results in much noisier examples. To do so, we compare the predicted transcription of an input signal with the transcription of its filtered version using metrics like WER and CER. We employed two filtering methods: a low-pass filter (LPF) with a 7 kHz cutoff frequency, eliminating high-frequency components (Monson et al., 2014) and a PyTorch-based Spectral Gating (SG) (Sainburg, 2019; Sainburg et al., 2020), an audio-denoising algorithm that calculates noise thresholds for each frequency band and generates masks to suppress noise below these thresholds. We then tried to distinguish attacks from benign data based on the resulting WER and CER values. When contrasting the accuracy results in Tab. 5 for AEs that have not been tailored to the classifier type with those in Tab. 7 for adaptive AEs, SG proves highly effective in distinguishing between adversarial and benign data across most models. This is especially evident with the NN classifier, which consistently matches or even surpasses the accuracy achieved by the non-tailored AEs, leading to an average accuracy boost from 97.07% to 97.64%, a gain of 0.57%.

## 6  DISCUSSION & CONCLUSION

We have demonstrated that characteristics of the distribution over the output tokens can serve as features of binary classifiers, turning them into an effective tool for identifying targeted adversarial attacks against ASR systems. As an example of such characteristics, the mean (w.r.t. the distributions from different time steps) of the median of the probabilities holds the greatest discriminative power across different models. Even on challenging data, these characteristics allow us to distinguish adversarial examples from benign data with high reliability. Our empirical findings strongly support employing a combination of these characteristics either in an ensemble of simple Gaussian classifiers or as input to a neural network to yield the best performance. This approach showcases exceptional discriminative power across a variety of modern ASR systems trained on different language corpora. It will be interesting to evaluate if the use of these characteristics of output distributions can also serve as indicators of other pertinent aspects, such as speech quality and intelligibility, which is a target for future work.

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

# A   APPENDIX

## A.1   ADAPTIVE ATTACK—ADDITIONAL SETTINGS

For an adaptive attack, we construct a new loss, $l_k$ explained in detail in Section 4.

$$l_k(x, \delta, \hat{y}) = (1 - \alpha) \cdot l(x, \delta, \hat{y}) + \alpha \cdot l_s^c(x) \ .$$

We perform 1,000 iterations on 100 randomly selected examples from the adversarial dataset, beginning with inputs that already mislead the system. We evaluated the adaptive attacks resulting from different settings for the minimization procedure of the loss:

1. We kept $\alpha$ constant at 0.3, while $\delta$ remained only unchanged during the initial 500 iterations. Afterward, $\delta$ is gradually reduced each time the perturbed signal successfully deceives the system, as defined by the C&W attack.

2. We experimented with three fixed $\alpha$ values: 0.3, 0.6, and 0.9, while $\delta$ is gradually reduced each time the perturbed signal successfully deceives the system, as defined by the C&W attack.

3. We increased the $\alpha$ value by 20% after each successful attack, while the $\delta$ factor remains unchanged during the initial 30 iterations.

4. We kept $\alpha$ constant at 0.3 and set the $l_s$ for attacking an EM, as defined in Section 4. We employed an EM-based on two characteristics: the median mean and the mean KLD.

5. We kept $\alpha$ constant at 0.3, and we redefined the $l_s$ term from the loss $l_k$ as follows:

$$l_s(x, \hat{x}) = \sum_{t=1}^{T-1} |D_{\mathrm{KL}}(p_x^{(t)} \| p_x^{(t+1)}) - D_{\mathrm{KL}}(p_{\hat{x}}^{(t)} \| p_{\hat{x}}^{(t+1)})| \ . \tag{3}$$

   where $T$ represents the length of the output utterance, $x$ the benign example, and $\hat{x}$ its adversarial counterpart.

6. We kept $\alpha$ constant at 0.3, and to minimize the statistical distance from the benign data distribution, we calculated for the same time step the KLD between the output distribution given the benign data $x$ and its adversarial counterpart $\hat{x}$, then we set the $l_s$ term to:

$$l_s(x, \hat{x}) = \sum_{t=1}^{T} |D_{\mathrm{KL}}(p_x^{(t)} \| p_{\hat{x}}^{(t)})| \ . \tag{4}$$

Results for the second setting are reported in Tab. 8. Regardless of the chosen $\alpha$ value, the second setting is unable to produce robust adversarial samples and has only a minimal effect on our proposed defense. This is due to the faster reduction of $\delta$, making it harder to generate an adaptive AE with smaller perturbations. Similar outcomes are evident in the fifth and sixth settings, where the modified loss $l_s$ does not yield improvement, as illustrated in Tab. 11, and Tab. 12. In the third configuration, some models exhibit enhanced outcomes by diminishing the discriminative capability of our defense. Nevertheless, the adaptive AEs generated in this scenario are characterized by noise, as indicated in Tab. 9, with a low $\mathrm{SNR}_{Seg}$. The fourth setting presents a noise improvement with higher $\mathrm{SNR}_{Seg}$ values compared to prior settings, as shown in Tab. 10. However, detectors are still able to discriminate many AEs from benign data. We opt for the first setting in the main paper, as it substantially diminishes our defense's discriminative power across all models. But, this comes at the expense of generating noisy data, results are presented in Tab. 13. We observed that using an $\alpha$ value above 0.3 increases the difficulty of generating adaptive adversarial examples.

Table 8: Quality of 100 generated adaptive adversarial samples with different $\alpha$ values. Evaluating AUROC for all models with 100 samples from the clean test set and 100 adaptive C&W AEs.

| Model | Score-Characteristic | $\alpha = 0.3$ | | $\alpha = 0.6$ | | $\alpha = 0.9$ | |
|---|---|---|---|---|---|---|---|
| | | $\mathrm{SNR}_{Seg}$ | GC AUROC | $\mathrm{SNR}_{Seg}$ | GC AUROC | $\mathrm{SNR}_{Seg}$ | GC AUROC |
| LSTM (It) | Mean-Median | 17.5 | 0.9635 | 17.13 | 0.8882 | 17.4 | 0.964 |
| LSTM (En) | Mean-Median | 14.78 | 0.9875 | 14.78 | 0.9741 | 14.75 | 0.9735 |
| LSTM (En-LM) | Max-Max | 17.42 | 0.9869 | 17.39 | 0.9762 | 17.37 | 0.9567 |
| wav2vec (Ma) | Mean-Entropy | 22.18 | 0.9912 | 22.16 | 0.9849 | 22.13 | 0.9779 |
| wav2vec (Ge) | Max-Min | 20.27 | 0.9774 | 19.82 | 0.8516 | 20.55 | 0.9803 |
| Trf (Ma) | Median-Max | 31.69 | 0.9893 | 31.74 | 0.9891 | 31.8 | 0.977 |
| Trf (En) | Max-Median | 27.54 | 1.000 | 27.61 | 1.000 | 27.65 | 1.000 |

Table 9: Quality of 100 generated adaptive C&W attacks using an adapted $\alpha$ value. Evaluating AUROC for all models with 100 samples from the clean test set and 100 adaptive C&W AEs.

| Model | Score-Characteristic | WER/CER | SER | $\mathbf{SNR}_{Seg}$ | SNR | GC AUROC |
|---|---|---|---|---|---|---|
| LSTM (It) | Mean-Median | 0.84% | 3.00% | 6.64 | 27.29 | 0.4656 |
| LSTM (En) | Mean-Median | 0.20% | 1.00% | 9.51 | 24.2 | 0.5333 |
| LSTM (En-LM) | Max-Max | 0.40% | 1.00% | 12.91 | 29.77 | 0.7437 |
| wav2vec (Ma) | Mean-Entropy | 0.08% | 1.00% | 13.76 | 21.34 | 0.9146 |
| wav2vec (Ge) | Max-Min | 0.00% | 0.00% | 12.21 | 33.7 | 0.6666 |
| Trf (Ma) | Median-Max | 0.00% | 0.00% | 14.48 | 24.71 | 0.7857 |
| Trf (En) | Max-Median | 0.00% | 0.00% | 15.91 | 35.49 | 0.9835 |

Table 10: Quality of 100 generated adaptive C&W attacks, keeping $\alpha$ constant at 0.3 and setting the $l_s$ loss for attacking an EM using mean median and mean KLD characteristics. Evaluating AUROC for all models with 100 samples from the clean test set and 100 adaptive C&W AEs.

| Model | WER/CER | SER | $\mathbf{SNR}_{Seg}$ | SNR | GC AUROC |
|---|---|---|---|---|---|
| LSTM (It) | 0.84% | 3.00% | 16 | 38.77 | 0.8014 |
| LSTM (En) | 1.09% | 2.00% | 14.08 | 30.29 | 0.8509 |
| LSTM (En-LM) | 1.19% | 2.00% | 17.25 | 35.37 | 0.8701 |
| wav2vec (Ma) | 0.08% | 1.00% | 21.91 | 29.83 | 0.523 |
| wav2vec (Ge) | 0.00% | 0.00% | 17.61 | 41.65 | 0.4437 |
| Trf (Ma) | 0.00% | 0.00% | 27.58 | 35.52 | 0.5286 |
| Trf (En) | 0.00% | 0.00% | 22.84 | 39.82 | 0.7631 |

Table 11: Quality of 100 generated adaptive C&W attacks keeping $\alpha$ constant at 0.3, and using the $l_s$ defined in equation 3. Evaluating AUROC for all models with 100 samples from the clean test set and 100 adaptive C&W AEs.

| Model | WER/CER | SER | $\mathbf{SNR}_{Seg}$ | SNR | GC AUROC |
|---|---|---|---|---|---|
| LSTM (It) | 0.84% | 3.00% | 17.73 | 44.37 | 0.9976 |
| LSTM (En) | 1.09% | 2.00% | 14.91 | 33.29 | 0.9993 |
| LSTM (En-LM) | 1.19% | 2.00% | 17.5 | 36.45 | 0.957 |
| wav2vec (Ma) | 0.08% | 1.00% | 22.22 | 31.35 | 0.9904 |
| wav2vec (Ge) | 0.00% | 0.00% | 20.58 | 50.86 | 0.9982 |
| Trf (Ma) | 0.00% | 0.00% | 30.41 | 42.81 | 0.9921 |

Table 12: Quality of 100 generated adaptive C&W attacks keeping $\alpha$ constant at 0.3, and using the $l_s$ defined in equation 4 to minimize the statistical distance from the benign data distribution. Evaluating AUROC for all models with 100 samples from the clean test set and 100 adaptive C&W AEs.

| Model | WER/CER | SER | $\mathbf{SNR}_{Seg}$ | SNR | GC AUROC |
|---|---|---|---|---|---|
| LSTM (It) | 0.84% | 3.00% | 17.76 | 44.5 | 0.9978 |
| LSTM (En) | 1.09% | 2.00% | 14.91 | 33.29 | 0.9993 |
| LSTM (En-LM) | 1.19% | 2.00% | 17.5 | 36.46 | 0.9551 |
| wav2vec (Ma) | 0.08% | 1.00% | 22.22 | 31.35 | 0.9902 |
| wav2vec (Ge) | 0.00% | 0.00% | 20.58 | 50.86 | 0.9982 |
| Trf (Ma) | 0.00% | 0.00% | 28.29 | 35.09 | 0.7562 |
| Trf (En) | 0.00% | 0.00% | 24.78 | 43.55 | 0.995 |

Table 13: Quality of 100 generated adaptive C&W attacks keeping $\alpha$ constant at 0.3, while $\delta$ factor remains unchanged during the initial 500 iterations. Evaluating AUROC for all models with 100 samples from the clean test set and 100 adaptive C&W AEs.

| Model | Score-Characteristic | WER/CER | SER | $\mathbf{SNR}_{Seg}$ | SNR | GC AUROC |
|---|---|---|---|---|---|---|
| LSTM (It) | Mean-Median | 0.84% | 3.00% | -1.47 | 18.36 | 0.335 |
| LSTM (En) | Mean-Median | 0.30% | 1.00% | 0.23 | 14.01 | 0.295 |
| LSTM (En-LM) | Max-Max | 0.40% | 1.00% | 3.18 | 16.82 | 0.425 |
| wav2vec (Ma) | Mean-Entropy | 0.08% | 1.00% | -4.30 | 4.09 | 0.375 |
| wav2vec (Ge) | Max-Min | 0.00% | 0.00% | -12.96 | 10.88 | 0.255 |
| Trf (Ma) | Median-Max | 0.00% | 0.00% | -1.09 | 8.01 | 0.285 |
| Trf (En) | Max-Median | 0.00% | 0.00% | -0.19 | 14.69 | 0.25 |

## A.2 CHARACTERISTIC RANKING

For the GCs, we determine the best-performing characteristics by ranking them according to the average AUROC on a validation set across all models. This ranking, which is shown in Tab. 14, determines the choice of characteristics to utilize for the EMs, where we implement a majority voting technique, using a total of $T \in 3, 5, 7, 9$ GCs.

Table 14: Ranking GCs based on the mean AUROC across all models on a validation set, using 100 benign data and 100 C&W AEs. $^{()}$ indicates the characteristic employed within a specific EM.

| Score-Characteristic | Benign vs. C&W adversarial data | Score-Characteristic | Benign vs. C&W adversarial data |
|---|---|---|---|
| Mean-Median$^{(3,5,7,9)}$ | **0.9872** | Mean-KLD | 0.9162 |
| Mean-Entropy$^{(3,5,7,9)}$ | **0.9871** | Max-JSD | 0.8480 |
| Max-Entropy$^{(3,5,7,9)}$ | **0.9808** | Max-KLD | 0.8319 |
| Median-Entropy$^{(5,7,9)}$ | **0.9796** | Min-Median | 0.8242 |
| Max-Median$^{(5,7,9)}$ | **0.9759** | Max-Max | 0.7764 |
| Median-Max$^{(7,9)}$ | **0.9733** | Min-Min | 0.7751 |
| Mean-Max$^{(7,9)}$ | **0.9617** | Min-Entropy | 0.7703 |
| Mean-Min$^{(9)}$ | **0.9541** | Min-KLD | 0.7066 |
| Min-Max$^{(9)}$ | **0.9523** | Mean-JSD | 0.6717 |
| Median-Median | 0.9488 | Median-KLD | 0.6706 |
| Max-Min | 0.9365 | Min-JSD | 0.6669 |
| Median-Min | 0.9339 | Median-JSD | 0.6368 |

## A.3 COMPUTATIONAL OVERHEAD

The assessment involves measuring the overall duration the system requires to predict 100 audio clips, utilizing an NVIDIA A40 with a memory capacity of 48 GB, results are reported in Tab. 15.

Table 15: Computational overhead to predict 100 audio clips measured in seconds.

| Model | Elapsed time | GC With detector | GC Overhead | GC Avg. time per sample | NN With detector | NN Overhead | NN Avg. time per sample |
|---|---|---|---|---|---|---|---|
| LSTM (It) | 53.005 | 66.964 | 13.959 | 0.140 | 53.452 | 0.447 | 0.004 |
| LSTM (En) | 55.742 | 69.224 | 13.482 | 0.135 | 58.038 | 2.295 | 0.023 |
| LSTM (En-LM) | 46.358 | 59.604 | 13.246 | 0.132 | 48.252 | 1.894 | 0.019 |
| wav2vec (Ma) | 13.339 | 22.799 | 9.460 | 0.095 | 14.772 | 1.432 | 0.014 |
| wav2vec (Ge) | 13.736 | 14.426 | 0.689 | 0.007 | 14.992 | 1.255 | 0.013 |
| Trf (Ma) | 32.070 | 39.675 | 7.605 | 0.076 | 34.656 | 2.586 | 0.026 |
| Trf (En) | 63.460 | 79.182 | 15.723 | 0.157 | 66.671 | 3.211 | 0.032 |
| Avg. | 39.67 | 50.27 | 10.59 | 0.11 | 41.55 | 1.87 | 0.02 |

It is worth noting that GCs take more time than NNs due to the utilization of a NumPy function that operates on the CPU.

## A.4 PERFORMANCE INDICATORS OF ASR SYSTEMS

We used the following, standard performance indicators:

**WER** The word error rate, is given by

$$\text{WER} = 100 \cdot \frac{S + D + I}{N} \ ,$$

where $S$, $D$, and $I$ are the number of words that were substituted, deleted, and inserted, respectively. The reference text's total word count, or $N$ is set to the number of ground-truth labels of the original test sample, or to those of the malicious target transcription for the adversarial attack, depending on which method is being evaluated. We aim for a model that has a low WER on the original data, i.e., it recognizes the ground-truth transcript with the highest possible accuracy. From the attacker's standpoint, the aim is to minimize the WER as well, but relative to the target transcription.

**SER**    The sentence error rate, derives from calculating the unsuccessful ratio related to the ground-truth label of the original test sample or the malicious target transcription for the adversarial attack, respectively.

$$\text{SER} = 100 \cdot \frac{N_E}{N} \quad,$$

where $N_E$ is the number of audio clips that have at least one transcription error, and $N$ is the total number of examples.

**SNR**    As reported in Carlini & Wagner (2018), the degree of distortion introduced by a perturbation $\delta$ in decibels (dB) is described as follows:

$$dB(x) = \max_i 20 \cdot \log_{10}(x_i),$$
$$SNR = dB(x) - dB(\delta),$$

where $x$ represents the clean audio signal, a higher SNR indicates a lower level of added noise.

**SNR$_{Seg}$**    The Segmental Signal-to-Noise Ratio measures the adversarial noise energy in Decibels and considers the entire audio signal. To obtain it, the energy ratios are computed segment by segment, which better reflects human perception than the non-segmental version (Mermelstein, 1979). The results are then averaged:

$$\text{SNR}_{\text{Seg}} = \frac{10}{M} \cdot \sum_{m=0}^{M-1} \log_{10} \frac{\sum_{t=mN}^{mN+N-1} x(t)^2}{\sum_{t=mN}^{mN+N-1} \delta(t)^2} \quad,$$

where $M$ is the number of frames in a signal and $N$ is the frame length, $x$ represents the clean audio signal and $\delta$ the adversarial perturbation. Thus, a higher SNR$_{Seg}$ indicates less additional noise.

## A.5    SPEECHBRAIN RECIPES PERFORMANCE

Tab. 16 presents the outcomes obtained using various SpeechBrain recipes across all datasets. The results align with those reported by Ravanelli et al. (2021), where detailed hyperparameter information for these models can be found.

Table 16: Performance of the ASR systems on benign data, in terms of word and sentence error rate, on the full test sets. LM denotes the language model.

| Model | Data | Language | LM | # Utterances | Tokenizer | WER/ CER | SER |
|-------|------|----------|-----|--------------|-----------|----------|------|
| LSTM | CV-Corpus | Italian | ✗ | 12,444 | BPE | 17.78% | 69.68% |
| LSTM | Librispeech | English | ✗ | 2,620 | BPE | 4.24% | 42.44% |
| LSTM | Librispeech | English | ✓ | 2,620 | BPE | 2.91% | 32.06% |
| wav2vec | Aishell | Mandarin | ✓ | 7,176 | Bert-Char | 5.05% | 39.30% |
| wav2vec | CV-Corpus | German | ✗ | 15,415 | Char | 10.31% | 46.56% |
| Trf | Aishell | Mandarin | ✗ | 7,176 | BPE | 6.23% | 42.35% |
| Trf | Librispeech | English | ✓ | 2,620 | BPE | 2.21% | 26.03% |

## A.6    UNTARGETED ATTACKS

To expand the range of adversarial attacks, we explore three untargeted attacks: PGD, Genetic, and Kenansville. The primary objective is to achieve a high WER, in contrast to C&W and psychoacoustic attacks where the aim is to minimize WER. Each adversarial attack type is evaluated under distinct settings. Regarding PGD, the perturbation $\delta$ is limited to a predefined value $\epsilon$, calculated as $\epsilon = ||x||_2 / 10^{\frac{SNR}{20}}$. We experimented with SNR values of 10 and 25. For Kenansville, the perturbation $\delta$ is controlled by removing frequencies that have a magnitude below a certain threshold $\theta$, determined by scaling the power of a signal with an SNR factor given by $10^{\frac{-SNR}{10}}$. Subsequently, all frequencies that have a cumulative power spectral density smaller than $\theta$ are set to zero, and the reconstructed signal is formed using the remaining frequencies. Our experiments involve an SNR value of 10, 15, and 25. Similar to PGD and Kenansville, the smaller perturbation is associated with higher SNR values. In Genetic attacks, the settings vary based on the number of iterations, we experimented with 1,000 and 2,000 iterations. The outcomes are detailed in Tab. 17 and Tab. 18.

In the case of PGD, we choose an SNR value of 25, as it induces a WER exceeding 50% across all models and maintains a higher segmental SNR compared to using an SNR of 10. As for Kenansville, we opt for an SNR value of 10. It is the only setting that demonstrates a genuine threat to the system by yielding a higher WER, but at the cost of generating noisy data. Using, for example, an SNR of 25 does not lead to a substantial change in the WER compared to the model's performance with benign data, or by adding background noise, as indicated in Tab. 16. In the context of Genetic attacks, adjusting the number of iterations doesn't result in a significant difference, failing to substantially degrade the system's performance, as the attack could not achieve a WER exceeding 50% across all models.

Table 17: Quality of 100 generated PGD using an SNR of 10 and 25, and Genetic attacks with a total number of iterations of 1,000 and 2,000. Results are measured by the average performance of the ASR systems across all models w.r.t the true labels as well as the SNRs.

| Model | PGD: 10 − 25 | | | | Genetic: 1,000 - 2,000 | | | |
|---|---|---|---|---|---|---|---|---|
| | WER | SER | $SNR_{Seg}$ | SNR | WER | SER | $SNR_{Seg}$ | SNR |
| LSTM (It) | 119% - 121% | 100% - 100% | -7.64 - 7.39 | 11.38 - 25.76 | 39.9% - 41.6% | 81% - 83% | 3.67 - 3.04 | 35.13 - 35.13 |
| LSTM (En) | 107% - 95% | 100% - 100% | -0.12 - 15.13 | 12.62 - 25.91 | 22.9% - 24.5% | 86% - 85% | 6.58 - 6.49 | 33.59 - 33.59 |
| LSTM (En-LM) | 108% - 100% | 100% - 100% | 0.01 - 15.19 | 12.59 - 26.21 | 20.7% - 23.8% | 79% - 83% | 7.00 - 6.63 | 33.59 - 33.59 |
| wav2vec (Ma) | 120% - 90% | 100% - 100% | 4.56 - 20.09 | 10.38 - 23.68 | 36.6% - 36.2% | 94% - 94% | 6.21 - 6.24 | 23.74 - 23.74 |
| wav2vec (Ge) | 118% - 102% | 100% - 100% | -8.28 - 6.88 | 12.45 - 26.79 | 28.4% - 30.7% | 76% - 78% | 2.44 - 1.72 | 33.39 - 33.39 |
| Trf (Ma) | 128% - 126% | 100% - 100% | 4.35 - 19.49 | 12.15 - 26.41 | 44.1% - 44.1% | 96% - 96% | 4.36 - 4.36 | 23.74 - 23.74 |
| Trf (En) | 109% - 102% | 100% - 100% | -0.49 - 14.88 | 13.10 - 26.58 | 15.9% - 17.8% | 74% - 77% | 9.16 - 8.79 | 33.59 - 33.59 |

Table 18: Quality of 100 generated Kenansville attacks with an SNR of 10, 15, and 25, measured by the average performance of the ASR systems across all models w.r.t the true labels as well as the SNRs.

| Model | Kenansville 10 − 15 − 25 | | | |
|---|---|---|---|---|
| | WER | SER | $SNR_{Seg}$ | SNR |
| LSTM (It) | 73.19% - 46.38% - 21.01% | 95.00% - 83.00% - 63.00% | -6.10 - -1.37 - 7.94 | 6.32 - 10.58 - 20.48 |
| LSTM (En) | 49.85% - 20.95% - 7.33% | 85.00% - 65.00% - 42.00% | 1.32 - 6.06 - 15.38 | 7.40 - 12.42 - 23.28 |
| LSTM (En-LM) | 49.33% - 19.92% - 5.26% | 78.00% - 56.00% - 29.00% | 1.32 - 6.06 - 15.38 | 7.40 - 12.42 - 23.28 |
| wav2vec (Ma) | 62.41% - 22.04% - 5.13% | 99.00% - 69.00% - 35.00% | 6.18 - 11.12 - 20.43 | 6.33 - 10.80 - 21.28 |
| wav2vec (Ge) | 49.32% - 26.71% - 10.62% | 86.00% - 67.00% - 35.00% | -5.73 - -1.22 - 7.83 | 6.89 - 11.93 - 22.83 |
| Trf (Ma) | 73.84% - 43.99% - 8.49% | 98.00% - 88.00% - 38.00% | 6.18 - 11.12 - 20.43 | 6.33 - 10.80 - 21.28 |
| Trf (En) | 40.66% - 13.73% - 4.02% | 72.00% - 46.00% - 24.00% | 1.32 - 6.06 - 15.38 | 7.40 - 12.42 - 23.28 |

To assess the performance of our detectors, we performed tests on GCs constructed based on either the mean-entropy or mean-median characteristic score. Following this, we measured the AUROC across all types of attacks. For NNs, we employed the NN model trained on the C&W attack to evaluate its performance across several attacks. The presenting findings are reported in Tab. 19, Tab. 20, and Tab. 21.

The GC trained on the mean-median characteristic demonstrates superior performance across a range of attacks compared to the mean-entropy characteristic, particularly in instances of C&W, psychoacoustic, PGD, and Kenansville. In the Kenansville attack, our detectors reduced effectiveness when using an SNR of 15 and 25. However, in these cases, the WER impact is minimal and even approaches the performance observed with benign examples. Our detectors generally struggle against the Genetic attack, likely due to its limited impact on the WER.

Table 19: AUROC assessment to detect AEs using GCs and NNs across targeted attacks. Evaluation of GCs based on mean entropy and mean median.

| Model | C&W | | | Psychoacoustic | | |
|---|---|---|---|---|---|---|
| | TD | GC: Entropy-Median | NN | TD | GC: Entropy-Median | NN |
| LSTM (It) | 0.8923 | 0.987 - 0.998 | 0.9962 | 0.893 | 0.986 - 0.997 | 0.9963 |
| LSTM (En) | 0.9697 | 1.000 - 0.999 | 0.9992 | 0.970 | 1.000 - 0.999 | 0.9992 |
| LSTM (En-LM) | 0.9293 | 0.991 - 0.951 | 0.9903 | 0.942 | 0.992 - 0.956 | 0.9912 |
| wav2vec (Ma) | 0.9937 | 0.995 - 0.990 | 0.9937 | 0.994 | 0.992 - 0.990 | 0.9947 |
| wav2vec (Ge) | 0.9836 | 0.948 - 0.998 | 0.991 | 0.984 | 0.929 - 0.997 | 0.9889 |
| Trf (Ma) | 0.9828 | 0.990 - 0.989 | 0.9969 | 0.991 | 0.987 - 0.993 | 0.9941 |
| Trf (En) | 0.9828 | 1.000 - 1.000 | 1.000 | 0.990 | 0.999 - 1.000 | 0.9999 |

Table 20: AUROC assessment to detect AEs using GCs and NNs across PGD and Genetic attacks. Evaluation of GCs based on mean entropy and mean median characteristics. PGD using an SNR of 10 and 25. Genetic using 1,000 and 2,000 iterations.

| Model | TD | PGD: 10 & 25 GC: Entropy-Median | NN | TD | Genetic: 1,000 & 2,000 GC: Entropy-Median | NN |
|---|---|---|---|---|---|---|
| LSTM (It) | 0.724 - 0.715 | 0.917-0.916 & 0.934-0.939 | 0.935 - 0.939 | 0.525 - 0.544 | 0.577-0.592 & 0.560-0.579 | 0.689 - 0.681 |
| LSTM (En) | 0.735 - 0.687 | 0.990-0.973 & 0.942-0.960 | 0.992 - 0.980 | 0.525 - 0.529 | 0.587-0.510 & 0.606-0.497 | 0.676 - 0.685 |
| LSTM (En-LM) | 0.785 - 0.825 | 0.807-1.000 & 0.703-0.998 | 0.640 - 0.782 | 0.560 - 0.549 | 0.662-0.614 & 0.684-0.638 | 0.680 - 0.694 |
| wav2vec (Ma) | 0.854 - 0.774 | 0.864-0.903 & 0.819-0.837 | 0.908 - 0.838 | 0.667 - 0.648 | 0.648-0.713 & 0.569-0.666 | 0.757 - 0.725 |
| wav2vec (Ge) | 0.749 - 0.818 | 0.751-0.895 & 0.710-0.779 | 0.239 - 0.395 | 0.532 - 0.553 | 0.523-0.640 & 0.563-0.648 | 0.575 - 0.580 |
| Trf (Ma) | 0.858 - 0.859 | 0.998-0.869 & 0.996-0.762 | 0.943 - 0.907 | 0.585 - 0.585 | 0.748-0.648 & 0.748-0.648 | 0.841 - 0.841 |
| Trf (En) | 0.735 - 0.754 | 0.400-0.907 & 0.527-0.750 | 0.357 - 0.586 | 0.549 - 0.527 | 0.477-0.440 & 0.483-0.456 | 0.631 - 0.640 |

Table 21: AUROC assessment to detect Kenansville AEs using GCs and NNs. Evaluation of GCs based on mean entropy and mean median. Evaluating the attack with an SNR of 10, 15, and 25.

| Model | TD | Kenansville: 10 & 15 & 25 GC: Entropy-Median | NN |
|---|---|---|---|
| LSTM (It) | 0.684 - 0.598 - 0.525 | 0.885-0.825 & 0.761-0.724 & 0.537-0.561 | 0.921 - 0.856 - 0.600 |
| LSTM (En) | 0.762 - 0.633 - 0.512 | 0.867-0.888 & 0.714-0.722 & 0.522-0.495 | 0.894 - 0.783 - 0.560 |
| LSTM (En-LM) | 0.762 - 0.647 - 0.508 | 0.866-0.788 & 0.700-0.620 & 0.525-0.511 | 0.887 - 0.710 - 0.514 |
| wav2vec (Ma) | 0.895 - 0.612 - 0.454 | 0.966-0.967 & 0.708-0.697 & 0.523-0.506 | 0.970 - 0.764 - 0.569 |
| wav2vec (Ge) | 0.774 - 0.690 - 0.538 | 0.750-0.927 & 0.564-0.794 & 0.473-0.591 | 0.851 - 0.683 - 0.554 |
| Trf (Ma) | 0.896 - 0.835 - 0.533 | 0.998-0.999 & 0.947-0.923 & 0.584-0.554 | 0.998 - 0.970 - 0.665 |
| Trf (En) | 0.731 - 0.619 - 0.496 | 0.851-0.894 & 0.628-0.654 & 0.512-0.493 | 0.943 - 0.772 - 0.542 |

## A.7 DETECTING PSYCHOACOUSTIC ATTACK

As described in Section 5.3, Tab. 22 present our classifiers' performance w.r.t. adversarial examples from the Psychoacoustic attack, and Tab. 23 reports the goodness-of-fit performance of our classifiers to Psychoacoustic AEs, aimed at maintaining an FPR below 1% (when applicable) and achieving a minimum TPR of 50%.

Table 22: Comparing classifiers on clean and noisy data, evaluating AUROC for all models using 100 samples from the clean test set and 100 Psychoacoustic AEs. [*] denotes best-performing score-characteristic.

| Model | Score-Characteristic[*] | Noisy vs. Psychoacoustic data TD | GC | NN | Benign vs. Psychoacoustic data TD | GC | NN |
|---|---|---|---|---|---|---|---|
| LSTM (It) | Mean-Median | 0.8411 | 0.9655 | 0.9537 | 0.8930 | 0.9972 | 0.9962 |
| LSTM (En) | Mean-Median | 0.9596 | 0.9969 | 0.9980 | 0.9699 | 0.9993 | 0.9990 |
| LSTM (En-LM) | Max-Max | 0.9172 | 0.9857 | 0.9849 | 0.9416 | 0.9851 | 0.9856 |
| wav2vec (Ma) | Mean-Entropy | 0.9812 | 0.9556 | 0.9668 | 0.9935 | 0.9915 | 0.9949 |
| wav2vec (Ge) | Max-Min | 0.9555 | 0.9464 | 0.9922 | 0.9835 | 0.9884 | 0.9959 |
| Trf (Ma) | Median-Max | 0.9868 | 0.9882 | 0.9743 | 0.9910 | 0.9980 | 0.9955 |
| Trf (En) | Max-Median | 0.9542 | 0.9245 | 0.9750 | 0.9903 | 0.9998 | 0.9999 |
| Average AUROC across all models | | 0.9422 | 0.9661 | **0.9778** | 0.9661 | 0.9942 | **0.9953** |

Table 23: Classification accuracies for classifiers, based on a threshold for a maximum 1% FPR (if possible) and a minimum 50% TPR, using 100 benign data and 100 Psychoacoustic AEs.

| Model | TD | GC | EM=3 | EM=5 | EM=7 | EM=9 | NN |
|---|---|---|---|---|---|---|---|
| LSTM-LAS-CTC-It | 72.50% / 0.05 | **98.00% / 0.01** | 94.50% / 0.01 | 92.50% / 0.01 | 90.00% / 0.01 | 91.50% / 0.00 | 95.00% / 0.01 |
| LSTM-LAS-CTC-En | 85.00% / 0.01 | 98.50% / 0.01 | **99.50% / 0.00** | **99.50% / 0.00** | **99.50% / 0.00** | **99.50% / 0.00** | 98.00% / 0.01 |
| LSTM-LAS-CTC-En-lm | 75.00% / 0.02 | 88.00% / 0.01 | 91.50% / 0.01 | 93.00% / 0.01 | 97.00% / 0.01 | 93.00% / 0.01 | 93.50% / 0.01 |
| wav2vec2-CTC-Ma | 96.50% / 0.01 | **97.50% / 0.01** | 97.50% / 0.01 | 96.00% / 0.00 | 90.50% / 0.00 | 92.50% / 0.00 | 97.00% / 0.00 |
| wav2vec2-CTC-Ge | 94.00% / 0.03 | 94.50% / 0.01 | 85.50% / 0.03 | 92.00% / 0.00 | 83.50% / 0.01 | 90.00% / 0.00 | **97.00% / 0.00** |
| Trf-LAS-CTC-Ma | 96.00% / 0.01 | **98.00% / 0.01** | 95.00% / 0.01 | 96.50% / 0.00 | 96.00% / 0.00 | 95.50% / 0.00 | 95.50% / 0.00 |
| Trf-LAS-CTC-En | 93.00% / 0.01 | 99.00% / 0.01 | 99.00% / 0.01 | **99.50% / 0.01** | **99.50% / 0.01** | **99.50% / 0.01** | 99.00% / 0.01 |
| Avg. accuracy / FPR | 87.43% / 0.02 | 96.21% / 0.01 | 94.64% / 0.01 | 95.57% / 0.00 | 93.71% / 0.01 | 94.50% / 0.00 | **96.43% / 0.01** |

## A.8 PERFORMANCE OF GAUSSIAN CLASSIFIERS

We build 24 Gaussian classifiers for each model based on 24 single scores. We then compare these GCs using both clean and noisy data, assessing AUROC and the area under the precision-recall curve (AUPRC) on a validation set; with 100 samples from the clean test dataset, 100 from C&W AEs, and 100 from Psychoacoustic AEs. The results are presented in the following sequence: Tab. 24 corresponds to the LSTM-LAS-CTC-It model, Tab. 25 corresponds to the LSTM-LAS-CTC-En model, Tab. 26 corresponds to the LSTM-LAS-CTC-En-lm model, Tab. 27 corresponds to the wav2vec2-CTC-Ma model, Tab. 28 corresponds to the wav2vec2-CTC-Ge model, Tab. 29 corresponds to the Trf-LAS-CTC-Ma model, Tab. 30 corresponds to the Trf-LAS-CTC-En model. The best AUROC value are shown in bold, as well as the top-performing score characteristic corresponding to the C&W attack.

Table 24: Comparing GCs on clean and noisy data for the LSTM-LAS-CTC-It model, assessing AUROC/AUPRC with 100 samples each from clean test data, C&W AEs, and Psychoacoustic AEs.

| GC Score-Characteristic | C&W attack | | Psychoacoustic attack | |
|---|---|---|---|---|
| | Noisy vs. adversarial data | Benign vs. adversarial data | Noisy vs. adversarial data | Benign vs. adversarial data |
| Mean-Entropy | 93.60% / 95.42% | 98.12% / 98.50% | 93.01% / 94.61% | 97.69% / 98.00% |
| Max-Entropy | 81.22% / 85.77% | 95.71% / 95.80% | 80.72% / 85.07% | 95.13% / 94.90% |
| Min-Entropy | 64.96% / 54.46% | 66.02% / 54.76% | 72.40% / 61.08% | 73.72% / 62.16% |
| Median-Entropy | 90.92% / 87.70% | 92.76% / 88.47% | 90.11% / 87.69% | 91.96% / 87.62% |
| Mean-Max | 92.88% / 94.14% | 96.28% / 97.09% | 92.14% / 92.44% | 95.56% / 95.35% |
| Max-Max | 73.95% / 65.94% | 74.26% / 66.06% | 74.36% / 66.37% | 74.82% / 66.59% |
| Min-Max | 80.64% / 83.77% | 90.81% / 90.77% | 79.98% / 82.85% | 89.84% / 89.24% |
| Median-Max | 89.30% / 84.59% | 90.27% / 82.59% | 88.03% / 82.54% | 89.11% / 81.68% |
| Mean-Min | 94.57% / 96.02% | 98.79% / 98.95% | 94.60% / 96.11% | **98.87% / 99.03%** |
| Max-Min | 80.90% / 86.15% | 97.06% / 97.22% | 80.84% / 86.29% | 97.20% / 97.36% |
| Min-Min | 75.82% / 72.01% | 74.92% / 69.92% | 76.75% / 72.28% | 75.65% / 70.16% |
| Median-Min | 92.85% / 95.22% | 98.55% / 98.83% | 92.92% / 95.33% | 98.60% / 98.88% |
| **Mean-Median** | **95.91% / 96.81%** | **98.87% / 99.00%** | **95.92% / 96.87%** | 98.87% / 99.02% |
| Max-Median | 84.01% / 88.29% | 96.21% / 96.98% | 83.32% / 87.82% | 96.02% / 96.87% |
| Min-Median | 82.40% / 78.14% | 80.03% / 74.35% | 80.50% / 73.84% | 78.83% / 71.63% |
| Median-Median | 93.87% / 95.73% | 98.05% / 98.51% | 93.95% / 95.85% | 98.15% / 98.60% |
| Mean-JSD | 44.35% / 48.13% | 49.17% / 53.87% | 43.34% / 46.66% | 48.06% / 53.75% |
| Max-JSD | 85.56% / 86.74% | 94.79% / 92.38% | 84.33% / 82.18% | 93.37% / 87.59% |
| Min-JSD | 57.27% / 48.97% | 55.89% / 49.16% | 55.54% / 48.03% | 54.05% / 48.18% |
| Median-JSD | 46.51% / 44.71% | 43.16% / 42.21% | 46.30% / 43.99% | 43.05% / 42.29% |
| Mean-KLD | 64.44% / 69.14% | 75.38% / 78.60% | 64.72% / 69.94% | 75.50% / 78.86% |
| Max-KLD | 69.96% / 69.59% | 79.28% / 78.90% | 70.62% / 72.10% | 79.67% / 79.53% |
| Min-KLD | 65.73% / 54.20% | 64.87% / 54.39% | 64.19% / 53.16% | 63.02% / 53.06% |
| Median-KLD | 40.87% / 41.23% | 40.29% / 41.31% | 41.52% / 41.45% | 40.65% / 41.47% |

Table 25: Comparing GCs on clean and noisy data for the LSTM-LAS-CTC-En model, assessing AUROC/AUPRC with 100 samples each from clean test data, C&W AEs, and Psychoacoustic AEs.

| GC Score-Characteristic | C&W attack | | Psychoacoustic attack | |
|---|---|---|---|---|
| | Noisy vs. adversarial data | Benign vs. adversarial data | Noisy vs. adversarial data | Benign vs. adversarial data |
| Mean-Entropy | **99.46% / 99.43%** | 99.79% / 99.79% | 99.51% / 99.49% | 99.81% / 99.81% |
| Max-Entropy | 98.32% / 98.33% | 99.45% / 99.39% | 98.47% / 98.55% | 99.60% / 99.59% |
| Min-Entropy | 69.62% / 61.50% | 67.92% / 61.43% | 68.84% / 61.19% | 66.86% / 60.79% |
| Median-Entropy | 99.28% / 99.38% | 99.09% / 99.46% | 99.45% / 99.54% | 99.13% / 99.50% |
| Mean-Max | 98.19% / 95.59% | 98.54% / 96.25% | 98.24% / 97.06% | 98.63% / 97.44% |
| Max-Max | 68.23% / 61.31% | 68.55% / 61.75% | 68.47% / 61.43% | 68.81% / 61.84% |
| Min-Max | 96.18% / 95.51% | 98.05% / 96.56% | 96.23% / 94.97% | 98.08% / 95.77% |
| Median-Max | 99.20% / 99.30% | 99.00% / 99.37% | 99.33% / 99.43% | 99.07% / 99.43% |
| Mean-Min | 97.43% / 97.83% | 98.97% / 99.02% | 97.58% / 97.90% | 98.97% / 98.99% |
| Max-Min | 92.52% / 93.11% | 94.71% / 95.08% | 92.55% / 93.04% | 94.88% / 95.21% |
| Min-Min | 55.12% / 53.24% | 58.95% / 56.46% | 56.52% / 54.70% | 59.94% / 57.26% |
| Median-Min | 97.37% / 97.90% | 99.10% / 99.19% | 97.78% / 98.25% | 99.35% / 99.40% |
| **Mean-Median** | **99.81% / 99.82%** | **99.98% / 99.98%** | **99.82% / 99.83%** | **99.96% / 99.96%** |
| Max-Median | 98.83% / 99.03% | 99.94% / 99.94% | 98.79% / 98.99% | 99.92% / 99.92% |
| Min-Median | 69.64% / 70.68% | 71.00% / 71.43% | 67.19% / 68.84% | 68.54% / 69.41% |
| Median-Median | 99.28% / 99.37% | 99.81% / 99.80% | 99.20% / 99.29% | 99.74% / 99.71% |
| Mean-JSD | 93.06% / 94.07% | 93.12% / 94.35% | 93.05% / 94.08% | 93.14% / 94.40% |
| Max-JSD | 97.92% / 97.84% | 98.72% / 98.08% | 97.93% / 97.84% | 98.71% / 98.07% |
| Min-JSD | 88.13% / 78.33% | 88.55% / 80.71% | 87.15% / 78.47% | 87.75% / 81.33% |
| Median-JSD | 24.99% / 44.39% | 24.07% / 42.58% | 25.11% / 44.72% | 24.14% / 43.02% |
| Mean-KLD | 99.25% / 99.43% | 98.03% / 98.87% | 99.26% / 99.43% | 98.03% / 98.87% |
| Max-KLD | 80.43% / 78.57% | 78.32% / 78.30% | 77.90% / 72.12% | 75.80% / 72.18% |
| Min-KLD | 86.51% / 79.26% | 85.98% / 78.56% | 86.19% / 77.83% | 85.37% / 76.93% |
| Median-KLD | 29.70% / 48.02% | 25.91% / 46.39% | 29.60% / 48.06% | 25.71% / 46.43% |

Table 26: Comparing GCs on clean and noisy data for the LSTM-LAS-CTC-En-lm model, assessing AUROC/AUPRC with 100 samples each from clean test data, C&W AEs, and Psychoacoustic AEs.

| GC Score-Characteristic | C&W attack | | Psychoacoustic attack | |
| --- | --- | --- | --- | --- |
| | Noisy vs. adversarial data | Benign vs. adversarial data | Noisy vs. adversarial data | Benign vs. adversarial data |
| Mean-Entropy | 98.89% / 99.23% | 98.95% / 99.31% | 98.99% / 99.27% | 99.06% / 99.36% |
| Max-Entropy | 96.04% / 95.74% | 97.75% / 97.27% | 95.87% / 95.45% | 97.45% / 96.82% |
| Min-Entropy | 99.22% / 99.35% | 99.13% / 99.25% | 99.08% / 99.23% | 98.95% / 99.08% |
| Median-Entropy | 98.46% / 99.03% | 99.35% / 99.50% | 98.58% / 99.08% | **99.43% / 99.55%** |
| Mean-Max | 97.60% / 98.38% | 98.50% / 99.14% | 97.75% / 98.46% | 98.58% / 99.17% |
| **Max-Max** | **99.49% / 99.54%** | **99.46% / 99.50%** | **99.39% / 99.46%** | 99.36% / 99.41% |
| Min-Max | 93.61% / 90.53% | 96.06% / 92.45% | 93.16% / 88.20% | 95.43% / 89.89% |
| Median-Max | 98.29% / 98.91% | 98.67% / 99.21% | 98.43% / 98.99% | 98.77% / 99.25% |
| Mean-Min | 73.03% / 74.13% | 73.14% / 74.06% | 74.05% / 74.86% | 74.05% / 74.80% |
| Max-Min | 65.05% / 63.48% | 67.34% / 66.45% | 64.44% / 60.30% | 66.39% / 64.77% |
| Min-Min | 72.77% / 72.42% | 71.40% / 68.60% | 72.93% / 72.33% | 71.40% / 68.17% |
| Median-Min | 60.71% / 59.09% | 60.65% / 57.99% | 61.52% / 59.08% | 61.37% / 57.98% |
| Mean-Median | 96.03% / 96.48% | 95.57% / 95.98% | 96.31% / 96.69% | 96.01% / 96.29% |
| Max-Median | 92.94% / 92.10% | 95.19% / 94.81% | 93.11% / 92.65% | 95.37% / 95.24% |
| Min-Median | 88.21% / 88.76% | 86.50% / 86.93% | 87.26% / 86.57% | 85.38% / 84.77% |
| Median-Median | 74.48% / 73.29% | 74.55% / 73.65% | 76.36% / 76.68% | 76.51% / 76.86% |
| Mean-JSD | 98.08% / 98.66% | 98.45% / 99.03% | 98.18% / 98.74% | 98.53% / 99.09% |
| Max-JSD | 97.06% / 97.27% | 97.56% / 97.42% | 96.68% / 96.43% | 97.17% / 96.68% |
| Min-JSD | 61.86% / 58.69% | 64.81% / 57.07% | 63.04% / 58.82% | 65.87% / 57.35% |
| Median-JSD | 98.48% / 99.01% | 98.75% / 99.20% | 98.55% / 99.04% | 98.73% / 99.20% |
| Mean-KLD | 98.36% / 99.09% | 98.66% / 99.22% | 98.40% / 99.10% | 98.70% / 99.24% |
| Max-KLD | 80.95% / 77.12% | 81.65% / 75.15% | 81.02% / 77.34% | 81.83% / 76.04% |
| Min-KLD | 83.43% / 85.42% | 88.07% / 88.05% | 83.16% / 85.20% | 87.87% / 87.53% |
| Median-KLD | 98.08% / 98.75% | 98.73% / 99.19% | 98.26% / 98.92% | 98.87% / 99.32% |

Table 27: Comparing GCs on clean and noisy data for the wav2vec2-CTC-Ma model, assessing AUROC/AUPRC with 100 samples each from clean test data, C&W AEs, and Psychoacoustic AEs.

| GC Score-Characteristic | C&W attack | | Psychoacoustic attack | |
| --- | --- | --- | --- | --- |
| | Noisy vs. adversarial data | Benign vs. adversarial data | Noisy vs. adversarial data | Benign vs. adversarial data |
| **Mean-Entropy** | 93.04% / 95.16% | **98.47% / 98.61%** | 93.59% / 95.63% | **98.77% / 98.90%** |
| Max-Entropy | 91.91% / 94.64% | 96.56% / 97.57% | 91.68% / 94.49% | 96.51% / 97.51% |
| Min-Entropy | 47.55% / 49.91% | 48.01% / 49.81% | 47.38% / 49.91% | 47.86% / 49.84% |
| Median-Entropy | 86.18% / 81.51% | 87.42% / 80.76% | 86.02% / 82.84% | 87.25% / 82.93% |
| Mean-Max | 80.18% / 74.73% | 85.62% / 77.48% | 82.05% / 76.15% | 87.77% / 79.78% |
| Max-Max | 46.55% / 49.28% | 47.10% / 50.65% | 48.81% / 49.87% | 49.35% / 51.46% |
| **Min-Max** | **94.13% / 94.88%** | 97.52% / 97.00% | 93.41% / 94.04% | 96.86% / 96.23% |
| Median-Max | 69.32% / 63.98% | 70.52% / 61.90% | 67.79% / 64.09% | 69.10% / 61.98% |
| Mean-Min | 80.56% / 76.94% | 86.24% / 81.21% | 80.98% / 78.34% | 86.65% / 83.25% |
| Max-Min | 89.44% / 90.26% | 95.87% / 94.33% | 90.44% / 92.36% | 96.88% / 96.71% |
| Min-Min | 44.87% / 47.62% | 44.48% / 48.22% | 46.93% / 49.43% | 46.34% / 49.11% |
| Median-Min | 88.04% / 87.10% | 91.44% / 89.60% | 88.89% / 89.11% | 92.15% / 91.85% |
| Mean-Median | 93.83% / 95.26% | 98.14% / 98.20% | 94.31% / 95.75% | 98.59% / 98.66% |
| Max-Median | 87.83% / 87.95% | 91.19% / 89.54% | 89.33% / 91.80% | 92.72% / 93.85% |
| Min-Median | 73.22% / 67.11% | 75.58% / 70.82% | 72.64% / 67.31% | 75.04% / 70.58% |
| Median-Median | 87.05% / 84.71% | 89.81% / 86.85% | 88.70% / 87.70% | 91.42% / 89.31% |
| Mean-JSD | 80.66% / 76.71% | 81.21% / 76.49% | 78.88% / 71.28% | 79.04% / 70.22% |
| Max-JSD | 92.95% / 87.95% | 94.22% / 89.17% | 93.57% / 90.23% | 94.97% / 91.10% |
| Min-JSD | 57.26% / 55.91% | 62.05% / 59.30% | 59.66% / 56.52% | 64.35% / 60.42% |
| Median-JSD | 80.90% / 79.42% | 77.87% / 77.49% | 80.08% / 78.89% | 76.91% / 76.89% |
| Mean-KLD | 92.26% / 89.71% | 95.05% / 91.98% | 92.40% / 90.90% | 95.33% / 93.27% |
| Max-KLD | 69.09% / 64.16% | 76.39% / 71.98% | 70.57% / 65.66% | 77.48% / 73.35% |
| Min-KLD | 57.10% / 55.29% | 61.31% / 57.78% | 59.26% / 57.74% | 63.30% / 59.76% |
| Median-KLD | 87.11% / 86.61% | 86.85% / 87.33% | 87.59% / 88.24% | 87.35% / 88.85% |

Table 28: Comparing GCs on clean and noisy data for the wav2vec2-CTC-Ge model, assessing AUROC/AUPRC with 100 samples each from clean test data, C&W AEs, and Psychoacoustic AEs.

| GC Score-Characteristic | C&W attack | | Psychoacoustic attack | |
|---|---|---|---|---|
| | Noisy vs. adversarial data | Benign vs. adversarial data | Noisy vs. adversarial data | Benign vs. adversarial data |
| Mean-Entropy | 96.59% / 96.09% | 97.82% / 97.55% | 93.94% / 93.13% | 95.77% / 95.06% |
| Max-Entropy | 95.87% / 96.82% | 98.32% / 98.67% | 93.14% / 91.81% | 96.12% / 93.33% |
| Min-Entropy | 65.23% / 58.29% | 66.77% / 60.20% | 68.85% / 63.13% | 70.70% / 64.21% |
| Median-Entropy | 97.07% / 96.74% | 97.82% / 97.20% | 95.92% / 94.82% | 97.13% / 95.34% |
| Mean-Max | 90.00% / 89.61% | 91.76% / 91.78% | 87.29% / 86.50% | 89.58% / 89.15% |
| Max-Max | 64.18% / 58.44% | 65.21% / 59.04% | 65.18% / 59.15% | 66.21% / 59.75% |
| Min-Max | 92.77% / 94.22% | 93.13% / 95.28% | 89.33% / 87.56% | 90.03% / 88.66% |
| Median-Max | 96.52% / 94.83% | 97.22% / 95.22% | 95.71% / 94.13% | 96.97% / 94.93% |
| Mean-Min | 98.42% / 98.78% | 99.16% / 99.51% | 96.61% / 96.73% | 98.49% / 98.74% |
| **Max-Min** | 96.53% / 97.25% | **99.55% / 99.68%** | 94.30% / 95.15% | 98.71% / 98.86% |
| Min-Min | 67.02% / 61.91% | 77.55% / 75.27% | 65.30% / 60.29% | 76.52% / 73.77% |
| Median-Min | 98.20% / 98.67% | 99.07% / 99.49% | 96.32% / 96.68% | 98.63% / 99.02% |
| Mean-Median | **98.82% / 99.16%** | 99.25% / 99.57% | 97.62% / 97.96% | **99.04% / 99.36%** |
| Max-Median | 97.22% / 97.86% | 99.45% / 99.63% | 95.12% / 95.87% | 98.86% / 99.09% |
| Min-Median | 70.49% / 69.20% | 78.55% / 74.29% | 69.07% / 63.69% | 75.94% / 67.67% |
| Median-Median | 98.71% / 99.06% | 99.20% / 99.54% | 97.39% / 97.77% | 99.01% / 99.37% |
| Mean-JSD | 58.86% / 56.90% | 59.26% / 59.52% | 59.25% / 58.03% | 59.55% / 59.80% |
| Max-JSD | 64.57% / 54.88% | 65.00% / 54.70% | 58.96% / 51.52% | 59.40% / 51.39% |
| Min-JSD | 40.62% / 42.72% | 42.76% / 43.14% | 37.17% / 40.86% | 39.20% / 41.35% |
| Median-JSD | 46.32% / 48.11% | 44.63% / 46.90% | 48.33% / 48.44% | 47.14% / 47.42% |
| Mean-KLD | 98.48% / 98.81% | 98.42% / 98.82% | **97.69% / 98.05%** | 97.58% / 98.02% |
| Max-KLD | 86.47% / 81.46% | 89.45% / 83.32% | 89.66% / 89.50% | 92.99% / 92.18% |
| Min-KLD | 41.15% / 42.52% | 45.35% / 46.92% | 38.02% / 41.01% | 41.60% / 43.86% |
| Median-KLD | 71.56% / 78.53% | 58.32% / 70.04% | 70.28% / 72.22% | 57.87% / 64.48% |

Table 29: Comparing GCs on clean and noisy data for the Trf-LAS-CTC-Ma model, assessing AUROC/AUPRC with 100 samples each from clean test data, C&W AEs, and Psychoacoustic AEs.

| GC Score-Characteristic | C&W attack | | Psychoacoustic attack | |
|---|---|---|---|---|
| | Noisy vs. adversarial data | Benign vs. adversarial data | Noisy vs. adversarial data | Benign vs. adversarial data |
| Mean-Entropy | 96.40% / 96.43% | 98.45% / 98.06% | 96.52% / 97.23% | **99.02% / 99.13%** |
| Max-Entropy | 94.49% / 94.62% | 97.14% / 96.86% | 94.05% / 94.70% | 97.10% / 97.04% |
| Min-Entropy | 72.69% / 67.68% | 76.23% / 67.65% | 76.59% / 73.99% | 80.11% / 74.27% |
| Median-Entropy | **97.58% / 95.16%** | 98.88% / 96.51% | **97.68% / 96.72%** | 99.00% / 97.56% |
| Mean-Max | 93.70% / 93.83% | 96.40% / 95.96% | 93.50% / 94.59% | 96.58% / 96.97% |
| Max-Max | 69.42% / 63.29% | 74.44% / 66.72% | 68.95% / 62.58% | 73.62% / 65.87% |
| Min-Max | 93.92% / 92.87% | 96.16% / 94.56% | 92.79% / 92.54% | 95.32% / 94.20% |
| **Median-Max** | 97.54% / 96.86% | **99.02% / 98.00%** | 97.49% / 96.30% | 98.98% / 97.20% |
| Mean-Min | 95.74% / 96.04% | 98.77% / 98.46% | 96.17% / 96.70% | 98.98% / 98.95% |
| Max-Min | 93.12% / 93.81% | 97.34% / 96.94% | 92.72% / 94.15% | 97.72% / 97.82% |
| Min-Min | 85.24% / 82.71% | 88.82% / 85.63% | 86.86% / 84.86% | 90.13% / 86.73% |
| Median-Min | 93.04% / 94.14% | 97.44% / 97.55% | 93.09% / 94.20% | 97.43% / 97.66% |
| Mean-Median | 95.01% / 95.26% | 98.13% / 97.78% | 95.52% / 95.94% | 98.38% / 98.26% |
| Max-Median | 91.35% / 90.84% | 92.88% / 91.94% | 90.56% / 91.23% | 92.21% / 92.43% |
| Min-Median | 82.69% / 80.71% | 86.73% / 83.25% | 85.16% / 82.30% | 89.08% / 85.54% |
| Median-Median | 91.26% / 92.40% | 95.82% / 95.87% | 91.55% / 92.59% | 96.00% / 96.10% |
| Mean-JSD | 52.98% / 52.20% | 54.71% / 53.46% | 53.21% / 52.81% | 54.48% / 54.20% |
| Max-JSD | 73.53% / 66.82% | 74.15% / 65.70% | 83.01% / 76.46% | 84.50% / 76.49% |
| Min-JSD | 90.71% / 83.49% | 90.96% / 83.14% | 82.17% / 74.58% | 82.22% / 72.62% |
| Median-JSD | 94.45% / 94.53% | 96.53% / 95.51% | 94.07% / 93.05% | 96.01% / 93.55% |
| Mean-KLD | 80.82% / 79.41% | 84.47% / 84.13% | 80.27% / 76.63% | 84.15% / 81.26% |
| Max-KLD | 73.86% / 66.19% | 75.07% / 68.79% | 73.29% / 65.43% | 74.50% / 68.51% |
| Min-KLD | 88.20% / 86.28% | 89.43% / 87.20% | 79.20% / 76.64% | 80.91% / 77.29% |
| Median-KLD | 94.43% / 94.82% | 96.89% / 95.78% | 93.64% / 91.61% | 96.37% / 93.15% |

Table 30: Comparing GCs on clean and noisy data for the Trf-LAS-CTC-En model, assessing AU-ROC/AUPRC with 100 samples each from clean test data, C&W AEs, and Psychoacoustic AEs.

| GC Score-Characteristic | C&W attack | | Psychoacoustic attack | |
|---|---|---|---|---|
| | Noisy vs. adversarial data | Benign vs. adversarial data | Noisy vs. adversarial data | Benign vs. adversarial data |
| Mean-Entropy | **98.97% / 99.11%** | 99.99% / 99.99% | 97.94% / 97.96% | 99.59% / 99.51% |
| Max-Entropy | 95.09% / 96.63% | 99.89% / 99.89% | 93.97% / 93.80% | 98.95% / 97.00% |
| Min-Entropy | 93.52% / 91.42% | 96.35% / 92.72% | 92.71% / 91.84% | 96.29% / 94.21% |
| Median-Entropy | 98.46% / 98.85% | 99.97% / 99.97% | 97.89% / 98.36% | 99.84% / 99.85% |
| Mean-Max | 98.61% / 98.80% | 99.94% / 99.94% | 96.72% / 96.35% | 98.84% / 98.44% |
| Max-Max | 92.48% / 90.88% | 96.33% / 94.28% | 91.38% / 89.91% | 96.19% / 93.95% |
| Min-Max | 92.52% / 90.97% | 96.82% / 94.17% | 90.32% / 88.30% | 94.86% / 91.34% |
| Median-Max | 98.35% / 98.78% | 99.92% / 99.92% | 97.87% / 98.34% | 99.83% / 99.84% |
| Mean-Min | 97.33% / 98.08% | 99.85% / 99.87% | 96.66% / 97.61% | 99.79% / 99.83% |
| Max-Min | 90.05% / 94.25% | 99.99% / 99.99% | 90.09% / 94.22% | 99.99% / 99.99% |
| Min-Min | 90.54% / 89.08% | 93.38% / 91.83% | 89.48% / 86.17% | 92.48% / 89.41% |
| Median-Min | 97.17% / 97.87% | 99.84% / 99.86% | 96.18% / 97.09% | 99.75% / 99.80% |
| Mean-Median | 98.64% / 99.00% | 99.98% / 99.98% | 98.21% / 98.67% | 99.93% / 99.93% |
| **Max-Median** | 92.99% / 95.66% | **100.00% / 100.00%** | 92.69% / 95.41% | **100.00% / 100.00%** |
| Min-Median | 93.07% / 92.56% | 95.60% / 95.16% | 92.30% / 91.37% | 94.98% / 94.35% |
| Median-Median | 98.97% / 99.19% | 99.99% / 99.99% | 98.60% / 98.92% | 99.96% / 99.96% |
| Mean-JSD | 57.15% / 58.76% | 56.21% / 56.74% | 59.34% / 59.40% | 58.64% / 58.36% |
| Max-JSD | 96.64% / 96.57% | 98.40% / 97.56% | 96.60% / 96.55% | 98.51% / 97.61% |
| Min-JSD | 81.69% / 75.06% | 81.12% / 73.62% | 79.60% / 70.53% | 78.99% / 68.75% |
| Median-JSD | 91.80% / 87.57% | 94.00% / 87.69% | 88.28% / 80.01% | 90.39% / 80.93% |
| Mean-KLD | 86.12% / 90.11% | 87.97% / 91.04% | 86.83% / 90.56% | 88.68% / 91.65% |
| Max-KLD | 87.34% / 88.25% | 89.12% / 89.82% | 88.33% / 88.08% | 89.86% / 89.47% |
| Min-KLD | 71.15% / 66.31% | 75.57% / 67.19% | 69.37% / 67.44% | 74.63% / 69.02% |
| Median-KLD | 88.82% / 80.65% | 90.94% / 80.30% | 86.70% / 79.41% | 88.63% / 77.89% |

## A.9 GOODNESS OF FIT IN BINARY CLASSIFICATION

To evaluate the performance of our classifiers, we compute various goodness-of-fit metrics, including accuracy, false positive rate (FPR), true positive rate (TPR), precision, recall, and F1 score. These metrics are derived from the analysis of two types of errors: false positives (FP) and true negatives (TN) across all models. To calculate these metrics, we employ a conservative threshold to achieve a maximum 1% FPR (if applicable) while maintaining a minimum 50% TPR.

**LSTM-LAS-CTC-It model performance** Tab. 31: C&W AEs vs. benign data, Tab. 32: C&W AEs vs. noisy data, Tab. 33: Psychoacoustic AEs vs. benign data, Tab. 34: Psychoacoustic AEs vs. noisy data.

Table 31: LSTM-LAS-CTC-It binary classifiers' goodness-of-fit metrics, using a threshold of maximum 1% FPR (if available) and a minimum 50% TPR, with 100 benign data and 100 C&W AEs.

| Classifier | Accuracy | TP | FP | TN | FN | FPR | TPR | Precision | Recall | F1 |
|---|---|---|---|---|---|---|---|---|---|---|
| TD | 72.50% | 50 | 5 | 95 | 50 | 0.05 | 0.50 | 0.91 | 0.50 | 0.65 |
| GC | 98.00% | 97 | 1 | 99 | 3 | 0.01 | 0.97 | 0.99 | 0.97 | 0.98 |
| EM=3 | 94.00% | 89 | 1 | 99 | 11 | 0.01 | 0.89 | 0.99 | 0.89 | 0.94 |
| EM=5 | 92.50% | 86 | 1 | 99 | 14 | 0.01 | 0.86 | 0.99 | 0.86 | 0.92 |
| EM=7 | 90.50% | 82 | 1 | 99 | 18 | 0.01 | 0.82 | 0.99 | 0.82 | 0.90 |
| EM=9 | 91.50% | 84 | 1 | 99 | 16 | 0.01 | 0.84 | 0.99 | 0.84 | 0.91 |
| NN | 95.00% | 91 | 1 | 99 | 9 | 0.01 | 0.91 | 0.99 | 0.91 | 0.95 |

Table 32: LSTM-LAS-CTC-It binary classifiers' goodness-of-fit metrics, using a threshold of maximum 1% FPR (if applicable) and a minimum 50% TPR, with 100 noisy data and 100 C&W AEs.

| Classifier | Accuracy | TP | FP | TN | FN | FPR | TPR | Precision | Recall | F1 |
|---|---|---|---|---|---|---|---|---|---|---|
| TD | 69.50% | 50 | 11 | 89 | 50 | 0.11 | 0.50 | 0.82 | 0.50 | 0.62 |
| GC | 92.00% | 85 | 1 | 99 | 15 | 0.01 | 0.85 | 0.99 | 0.85 | 0.91 |
| EM=3 | 84.50% | 70 | 1 | 99 | 30 | 0.01 | 0.70 | 0.99 | 0.70 | 0.82 |
| EM=5 | 82.00% | 65 | 1 | 99 | 35 | 0.01 | 0.65 | 0.98 | 0.65 | 0.78 |
| EM=7 | 81.00% | 63 | 1 | 99 | 37 | 0.01 | 0.63 | 0.98 | 0.63 | 0.77 |
| EM=9 | 82.00% | 65 | 1 | 99 | 35 | 0.01 | 0.65 | 0.98 | 0.65 | 0.78 |
| NN | 84.50% | 70 | 1 | 99 | 30 | 0.01 | 0.70 | 0.99 | 0.70 | 0.82 |

Table 33: LSTM-LAS-CTC-It binary classifiers' goodness-of-fit metrics, with a threshold of maximum 1% FPR (if exist) and a minimum 50% TPR, with 100 benign and 100 Psychoacoustic AEs.

| Classifier | Accuracy | TP | FP | TN | FN | FPR | TPR | Precision | Recall | F1 |
|---|---|---|---|---|---|---|---|---|---|---|
| TD | 72.50% | 50 | 5 | 95 | 50 | 0.05 | 0.50 | 0.91 | 0.50 | 0.65 |
| GC | 98.00% | 97 | 1 | 99 | 3 | 0.01 | 0.97 | 0.99 | 0.97 | 0.98 |
| EM=3 | 94.50% | 90 | 1 | 99 | 10 | 0.01 | 0.90 | 0.99 | 0.90 | 0.94 |
| EM=5 | 92.50% | 86 | 1 | 99 | 14 | 0.01 | 0.86 | 0.99 | 0.86 | 0.92 |
| EM=7 | 90.00% | 81 | 1 | 99 | 19 | 0.01 | 0.81 | 0.99 | 0.81 | 0.89 |
| EM=9 | 91.50% | 83 | 0 | 100 | 17 | 0.00 | 0.83 | 1.00 | 0.83 | 0.91 |
| NN | 95.00% | 91 | 1 | 99 | 9 | 0.01 | 0.91 | 0.99 | 0.91 | 0.95 |

Table 34: LSTM-LAS-CTC-It binary classifiers' goodness-of-fit metrics, with a threshold of maximum 1% FPR (if exist) and a minimum 50% TPR, with 100 noisy data and 100 Psychoacoustic AEs.

| Classifier | Accuracy | TP | FP | TN | FN | FPR | TPR | Precision | Recall | F1 |
|---|---|---|---|---|---|---|---|---|---|---|
| TD | 69.50% | 50 | 11 | 89 | 50 | 0.11 | 0.50 | 0.82 | 0.50 | 0.62 |
| GC | 92.00% | 85 | 1 | 99 | 15 | 0.01 | 0.85 | 0.99 | 0.85 | 0.91 |
| EM=3 | 85.00% | 71 | 1 | 99 | 29 | 0.01 | 0.71 | 0.99 | 0.71 | 0.83 |
| EM=5 | 82.50% | 66 | 1 | 99 | 34 | 0.01 | 0.66 | 0.99 | 0.66 | 0.79 |
| EM=7 | 81.50% | 64 | 1 | 99 | 36 | 0.01 | 0.64 | 0.98 | 0.64 | 0.78 |
| EM=9 | 81.50% | 64 | 1 | 99 | 36 | 0.01 | 0.64 | 0.98 | 0.64 | 0.78 |
| NN | 85.00% | 71 | 1 | 99 | 29 | 0.01 | 0.71 | 0.99 | 0.71 | 0.83 |

**LSTM-LAS-CTC-En model performance**  Tab. 35: C&W AEs vs. benign data, Tab. 36: C&W AEs vs. noisy data, Tab. 37: Psychoacoustic AEs vs. benign data, Tab. 38: Psychoacoustic AEs vs. noisy data.

Table 35: LSTM-LAS-CTC-En binary classifiers' goodness-of-fit metrics, using a threshold of maximum 1% FPR (if available) and a minimum 50% TPR, with 100 benign data and 100 C&W AEs.

| Classifier | Accuracy | TP | FP | TN | FN | FPR | TPR | Precision | Recall | F1 |
|---|---|---|---|---|---|---|---|---|---|---|
| TD | 85.00% | 70 | 1 | 100 | 29 | 0.01 | 0.71 | 0.99 | 0.71 | 0.82 |
| GC | 98.50% | 98 | 1 | 99 | 2 | 0.01 | 0.98 | 0.99 | 0.98 | 0.98 |
| EM=3 | 99.50% | 99 | 0 | 100 | 1 | 0.00 | 0.99 | 1.00 | 0.99 | 0.99 |
| EM=5 | 99.50% | 99 | 0 | 100 | 1 | 0.00 | 0.99 | 1.00 | 0.99 | 0.99 |
| EM=7 | 99.50% | 99 | 0 | 100 | 1 | 0.00 | 0.99 | 1.00 | 0.99 | 0.99 |
| EM=9 | 99.50% | 99 | 0 | 100 | 1 | 0.00 | 0.99 | 1.00 | 0.99 | 0.99 |
| NN | 98.50% | 97 | 0 | 100 | 3 | 0.00 | 0.97 | 1.00 | 0.97 | 0.98 |

Table 36: LSTM-LAS-CTC-En binary classifiers' goodness-of-fit metrics, using a threshold of maximum 1% FPR (if applicable) and a minimum 50% TPR, with 100 noisy data and 100 C&W AEs.

| Classifier | Accuracy | TP | FP | TN | FN | FPR | TPR | Precision | Recall | F1 |
|---|---|---|---|---|---|---|---|---|---|---|
| TD | 80.00% | 60 | 1 | 100 | 39 | 0.01 | 0.61 | 0.98 | 0.61 | 0.75 |
| GC | 97.00% | 95 | 1 | 99 | 5 | 0.01 | 0.95 | 0.99 | 0.95 | 0.97 |
| EM=3 | 96.50% | 93 | 0 | 100 | 7 | 0.00 | 0.93 | 1.00 | 0.93 | 0.96 |
| EM=5 | 97.00% | 94 | 0 | 100 | 6 | 0.00 | 0.94 | 1.00 | 0.94 | 0.97 |
| EM=7 | 97.50% | 95 | 0 | 100 | 5 | 0.00 | 0.95 | 1.00 | 0.95 | 0.97 |
| EM=9 | 97.50% | 95 | 0 | 100 | 5 | 0.00 | 0.95 | 1.00 | 0.95 | 0.97 |
| NN | 99.00% | 99 | 1 | 99 | 1 | 0.01 | 0.99 | 0.99 | 0.99 | 0.99 |

Table 37: LSTM-LAS-CTC-En binary classifiers' goodness-of-fit metrics, with a threshold of maximum 1% FPR (if exist) and a minimum 50% TPR, with 100 benign and 100 Psychoacoustic AEs.

| Classifier | Accuracy | TP | FP | TN | FN | FPR | TPR | Precision | Recall | F1 |
|---|---|---|---|---|---|---|---|---|---|---|
| TD | 85.00% | 70 | 1 | 100 | 29 | 0.01 | 0.71 | 0.99 | 0.71 | 0.82 |
| GC | 98.50% | 98 | 1 | 99 | 2 | 0.01 | 0.98 | 0.99 | 0.98 | 0.98 |
| EM=3 | 99.50% | 99 | 0 | 100 | 1 | 0.00 | 0.99 | 1.00 | 0.99 | 0.99 |
| EM=5 | 99.50% | 99 | 0 | 100 | 1 | 0.00 | 0.99 | 1.00 | 0.99 | 0.99 |
| EM=7 | 99.50% | 99 | 0 | 100 | 1 | 0.00 | 0.99 | 1.00 | 0.99 | 0.99 |
| EM=9 | 99.50% | 99 | 0 | 100 | 1 | 0.00 | 0.99 | 1.00 | 0.99 | 0.99 |
| NN | 98.00% | 97 | 1 | 99 | 3 | 0.01 | 0.97 | 0.99 | 0.97 | 0.98 |

Table 38: LSTM-LAS-CTC-En binary classifiers' goodness-of-fit metrics, with a threshold of maximum 1% FPR (if exist) and a minimum 50% TPR, with 100 noisy data and 100 Psychoacoustic AEs.

| Classifier | Accuracy | TP | FP | TN | FN | FPR | TPR | Precision | Recall | F1 |
|---|---|---|---|---|---|---|---|---|---|---|
| TD | 80.00% | 60 | 1 | 100 | 39 | 0.01 | 0.61 | 0.98 | 0.61 | 0.75 |
| GC | 98.50% | 98 | 1 | 99 | 2 | 0.01 | 0.98 | 0.99 | 0.98 | 0.98 |
| EM=3 | 96.50% | 93 | 0 | 100 | 7 | 0.00 | 0.93 | 1.00 | 0.93 | 0.96 |
| EM=5 | 97.00% | 94 | 0 | 100 | 6 | 0.00 | 0.94 | 1.00 | 0.94 | 0.97 |
| EM=7 | 97.50% | 95 | 0 | 100 | 5 | 0.00 | 0.95 | 1.00 | 0.95 | 0.97 |
| EM=9 | 97.50% | 95 | 0 | 100 | 5 | 0.00 | 0.95 | 1.00 | 0.95 | 0.97 |
| NN | 97.00% | 95 | 1 | 99 | 5 | 0.01 | 0.95 | 0.99 | 0.95 | 0.97 |

**LSTM-LAS-CTC-En-lm model performance**  Tab. 39: C&W AEs vs. benign data, Tab. 40: C&W AEs vs. noisy data, Tab. 41: Psychoacoustic AEs vs. benign data, Tab. 42: Psychoacoustic AEs vs. noisy data.

Table 39: LSTM-LAS-CTC-En-lm binary classifiers' goodness-of-fit metrics, using a threshold of maximum 1% FPR (if available) and a minimum 50% TPR, with 100 benign data and 100 C&W AEs.

| Classifier | Accuracy | TP | FP | TN | FN | FPR | TPR | Precision | Recall | F1 |
|---|---|---|---|---|---|---|---|---|---|---|
| TD | 74.00% | 50 | 2 | 98 | 50 | 0.02 | 0.50 | 0.96 | 0.50 | 0.66 |
| GC | 90.50% | 82 | 1 | 99 | 18 | 0.01 | 0.82 | 0.99 | 0.82 | 0.90 |
| EM=3 | 91.00% | 83 | 1 | 99 | 17 | 0.01 | 0.83 | 0.99 | 0.83 | 0.90 |
| EM=5 | 92.50% | 86 | 1 | 99 | 14 | 0.01 | 0.86 | 0.99 | 0.86 | 0.92 |
| EM=7 | 97.00% | 95 | 1 | 99 | 5 | 0.01 | 0.95 | 0.99 | 0.95 | 0.97 |
| EM=9 | 92.50% | 86 | 1 | 99 | 14 | 0.01 | 0.86 | 0.99 | 0.86 | 0.92 |
| NN | 95.00% | 91 | 1 | 99 | 9 | 0.01 | 0.91 | 0.99 | 0.91 | 0.95 |

Table 40: LSTM-LAS-CTC-En-lm binary classifiers' goodness-of-fit metrics, using a threshold of maximum 1% FPR (if applicable) and a minimum 50% TPR, with 100 noisy data and 100 C&W AEs.

| Classifier | Accuracy | TP | FP | TN | FN | FPR | TPR | Precision | Recall | F1 |
|---|---|---|---|---|---|---|---|---|---|---|
| TD | 73.00% | 50 | 4 | 96 | 50 | 0.04 | 0.50 | 0.93 | 0.50 | 0.65 |
| GC | 90.00% | 81 | 1 | 99 | 19 | 0.01 | 0.81 | 0.99 | 0.81 | 0.89 |
| EM=3 | 87.00% | 75 | 1 | 99 | 25 | 0.01 | 0.75 | 0.99 | 0.75 | 0.85 |
| EM=5 | 88.50% | 78 | 1 | 99 | 22 | 0.01 | 0.78 | 0.99 | 0.78 | 0.87 |
| EM=7 | 93.00% | 87 | 1 | 99 | 13 | 0.01 | 0.87 | 0.99 | 0.87 | 0.93 |
| EM=9 | 89.50% | 80 | 1 | 99 | 20 | 0.01 | 0.80 | 0.99 | 0.80 | 0.88 |
| NN | 92.50% | 86 | 1 | 99 | 14 | 0.01 | 0.86 | 0.99 | 0.86 | 0.92 |

Table 41: LSTM-LAS-CTC-En-lm binary classifiers' goodness-of-fit metrics, with a threshold of maximum 1% FPR (if exist) and a minimum 50% TPR, with 100 benign and 100 Psychoacoustic AEs.

| Classifier | Accuracy | TP | FP | TN | FN | FPR | TPR | Precision | Recall | F1 |
|---|---|---|---|---|---|---|---|---|---|---|
| TD | 75.00% | 52 | 2 | 98 | 48 | 0.02 | 0.52 | 0.96 | 0.52 | 0.68 |
| GC | 88.00% | 77 | 1 | 99 | 23 | 0.01 | 0.77 | 0.99 | 0.77 | 0.87 |
| EM=3 | 91.50% | 84 | 1 | 99 | 16 | 0.01 | 0.84 | 0.99 | 0.84 | 0.91 |
| EM=5 | 93.00% | 87 | 1 | 99 | 13 | 0.01 | 0.87 | 0.99 | 0.87 | 0.93 |
| EM=7 | 97.00% | 95 | 1 | 99 | 5 | 0.01 | 0.95 | 0.99 | 0.95 | 0.97 |
| EM=9 | 93.00% | 87 | 1 | 99 | 13 | 0.01 | 0.87 | 0.99 | 0.87 | 0.93 |
| NN | 93.50% | 88 | 1 | 99 | 12 | 0.01 | 0.88 | 0.99 | 0.88 | 0.93 |

Table 42: LSTM-LAS-CTC-En-lm binary classifiers' goodness-of-fit metrics, with a threshold of maximum 1% FPR (if exist) and a minimum 50% TPR, with 100 noisy data and 100 Psychoacoustic AEs.

| Classifier | Accuracy | TP | FP | TN | FN | FPR | TPR | Precision | Recall | F1 |
|---|---|---|---|---|---|---|---|---|---|---|
| TD | 74.50% | 52 | 3 | 97 | 48 | 0.03 | 0.52 | 0.95 | 0.52 | 0.67 |
| GC | 88.00% | 77 | 1 | 99 | 23 | 0.01 | 0.77 | 0.99 | 0.77 | 0.87 |
| EM=3 | 87.00% | 75 | 1 | 99 | 25 | 0.01 | 0.75 | 0.99 | 0.75 | 0.85 |
| EM=5 | 88.50% | 78 | 1 | 99 | 22 | 0.01 | 0.78 | 0.99 | 0.78 | 0.87 |
| EM=7 | 93.00% | 87 | 1 | 99 | 13 | 0.01 | 0.87 | 0.99 | 0.87 | 0.93 |
| EM=9 | 89.50% | 80 | 1 | 99 | 20 | 0.01 | 0.80 | 0.99 | 0.80 | 0.88 |
| NN | 94.00% | 89 | 1 | 99 | 11 | 0.01 | 0.89 | 0.99 | 0.89 | 0.94 |

**wav2vec2-CTC-Ma model performance**  Tab. 43: C&W AEs vs. benign data, Tab. 44: C&W AEs vs. noisy data, Tab. 45: Psychoacoustic AEs vs. benign data, Tab. 46: Psychoacoustic AEs vs. noisy data.

Table 43: wav2vec2-CTC-Ma binary classifiers' goodness-of-fit metrics, using a threshold of maximum 1% FPR (if available) and a minimum 50% TPR, with 100 benign data and 100 C&W AEs.

| Classifier | Accuracy | TP | FP | TN | FN | FPR | TPR | Precision | Recall | F1 |
|---|---|---|---|---|---|---|---|---|---|---|
| TD | 97.00% | 95 | 1 | 99 | 5 | 0.01 | 0.95 | 0.99 | 0.95 | 0.97 |
| GC | 97.50% | 96 | 1 | 99 | 4 | 0.01 | 0.96 | 0.99 | 0.96 | 0.97 |
| EM=3 | 97.50% | 96 | 1 | 99 | 4 | 0.01 | 0.96 | 0.99 | 0.96 | 0.97 |
| EM=5 | 96.00% | 92 | 0 | 100 | 8 | 0.00 | 0.92 | 1.00 | 0.92 | 0.96 |
| EM=7 | 91.50% | 83 | 0 | 100 | 17 | 0.00 | 0.83 | 1.00 | 0.83 | 0.91 |
| EM=9 | 93.50% | 87 | 0 | 100 | 13 | 0.00 | 0.87 | 1.00 | 0.87 | 0.93 |
| NN | 98.50% | 98 | 1 | 99 | 2 | 0.01 | 0.98 | 0.99 | 0.98 | 0.98 |

Table 44: wav2vec2-CTC-Ma binary classifiers' goodness-of-fit metrics, using a threshold of maximum 1% FPR (if applicable) and a minimum 50% TPR, with 100 noisy data and 100 C&W AEs.

| Classifier | Accuracy | TP | FP | TN | FN | FPR | TPR | Precision | Recall | F1 |
|---|---|---|---|---|---|---|---|---|---|---|
| TD | 69.50% | 50 | 11 | 89 | 50 | 0.11 | 0.50 | 0.82 | 0.50 | 0.62 |
| GC | 92.00% | 85 | 1 | 99 | 15 | 0.01 | 0.85 | 0.99 | 0.85 | 0.91 |
| EM=3 | 84.50% | 70 | 1 | 99 | 30 | 0.01 | 0.70 | 0.99 | 0.70 | 0.82 |
| EM=5 | 82.00% | 65 | 1 | 99 | 35 | 0.01 | 0.65 | 0.98 | 0.65 | 0.78 |
| EM=7 | 81.00% | 63 | 1 | 99 | 37 | 0.01 | 0.63 | 0.98 | 0.63 | 0.77 |
| EM=9 | 82.00% | 65 | 1 | 99 | 35 | 0.01 | 0.65 | 0.98 | 0.65 | 0.78 |
| NN | 84.50% | 70 | 1 | 99 | 30 | 0.01 | 0.70 | 0.99 | 0.70 | 0.82 |

Table 45: wav2vec2-CTC-Ma binary classifiers' goodness-of-fit metrics, with a threshold of maximum 1% FPR (if exist) and a minimum 50% TPR, with 100 benign and 100 Psychoacoustic AEs.

| Classifier | Accuracy | TP | FP | TN | FN | FPR | TPR | Precision | Recall | F1 |
|---|---|---|---|---|---|---|---|---|---|---|
| TD | 72.50% | 50 | 5 | 95 | 50 | 0.05 | 0.50 | 0.91 | 0.50 | 0.65 |
| GC | 98.00% | 97 | 1 | 99 | 3 | 0.01 | 0.97 | 0.99 | 0.97 | 0.98 |
| EM=3 | 94.50% | 90 | 1 | 99 | 10 | 0.01 | 0.90 | 0.99 | 0.90 | 0.94 |
| EM=5 | 92.50% | 86 | 1 | 99 | 14 | 0.01 | 0.86 | 0.99 | 0.86 | 0.92 |
| EM=7 | 90.00% | 81 | 1 | 99 | 19 | 0.01 | 0.81 | 0.99 | 0.81 | 0.89 |
| EM=9 | 91.50% | 83 | 0 | 100 | 17 | 0.00 | 0.83 | 1.00 | 0.83 | 0.91 |
| NN | 95.00% | 91 | 1 | 99 | 9 | 0.01 | 0.91 | 0.99 | 0.91 | 0.95 |

Table 46: wav2vec2-CTC-Ma binary classifiers' goodness-of-fit metrics, with a threshold of maximum 1% FPR (if exist) and a minimum 50% TPR, with 100 noisy data and 100 Psychoacoustic AEs.

| Classifier | Accuracy | TP | FP | TN | FN | FPR | TPR | Precision | Recall | F1 |
|---|---|---|---|---|---|---|---|---|---|---|
| TD | 69.50% | 50 | 11 | 89 | 50 | 0.11 | 0.50 | 0.82 | 0.50 | 0.62 |
| GC | 92.00% | 85 | 1 | 99 | 15 | 0.01 | 0.85 | 0.99 | 0.85 | 0.91 |
| EM=3 | 85.00% | 71 | 1 | 99 | 29 | 0.01 | 0.71 | 0.99 | 0.71 | 0.83 |
| EM=5 | 82.50% | 66 | 1 | 99 | 34 | 0.01 | 0.66 | 0.99 | 0.66 | 0.79 |
| EM=7 | 81.50% | 64 | 1 | 99 | 36 | 0.01 | 0.64 | 0.98 | 0.64 | 0.78 |
| EM=9 | 81.50% | 64 | 1 | 99 | 36 | 0.01 | 0.64 | 0.98 | 0.64 | 0.78 |
| NN | 85.00% | 71 | 1 | 99 | 29 | 0.01 | 0.71 | 0.99 | 0.71 | 0.83 |

**wav2vec2-CTC-Ge model performance** Tab. 47: C&W AEs vs. benign data, Tab. 48: C&W AEs vs. noisy data, Tab. 49: Psychoacoustic AEs vs. benign data, Tab. 50: Psychoacoustic AEs vs. noisy data.

Table 47: wav2vec2-CTC-Ge binary classifiers' goodness-of-fit metrics, using a threshold of maximum 1% FPR (if available) and a minimum 50% TPR, with 100 benign data and 100 C&W AEs.

| Classifier | Accuracy | TP | FP | TN | FN | FPR | TPR | Precision | Recall | F1 |
|---|---|---|---|---|---|---|---|---|---|---|
| TD | 94.00% | 91 | 3 | 97 | 9 | 0.03 | 0.91 | 0.97 | 0.91 | 0.94 |
| GC | 98.00% | 97 | 1 | 99 | 3 | 0.01 | 0.97 | 0.99 | 0.97 | 0.98 |
| EM=3 | 97.00% | 94 | 0 | 100 | 6 | 0.00 | 0.94 | 1.00 | 0.94 | 0.97 |
| EM=5 | 98.00% | 96 | 0 | 100 | 4 | 0.00 | 0.96 | 1.00 | 0.96 | 0.98 |
| EM=7 | 96.50% | 93 | 0 | 100 | 7 | 0.00 | 0.93 | 1.00 | 0.93 | 0.96 |
| EM=9 | 98.00% | 96 | 0 | 100 | 4 | 0.00 | 0.96 | 1.00 | 0.96 | 0.98 |
| NN | 97.00% | 95 | 1 | 99 | 5 | 0.01 | 0.95 | 0.99 | 0.95 | 0.97 |

Table 48: wav2vec2-CTC-Ge binary classifiers' goodness-of-fit metrics, using a threshold of maximum 1% FPR (if applicable) and a minimum 50% TPR, with 100 noisy data and 100 C&W AEs.

| Classifier | Accuracy | TP | FP | TN | FN | FPR | TPR | Precision | Recall | F1 |
|---|---|---|---|---|---|---|---|---|---|---|
| TD | 91.50% | 91 | 8 | 92 | 9 | 0.08 | 0.91 | 0.92 | 0.91 | 0.91 |
| GC | 92.50% | 86 | 1 | 99 | 14 | 0.01 | 0.86 | 0.99 | 0.86 | 0.92 |
| EM=3 | 92.50% | 85 | 0 | 100 | 15 | 0.00 | 0.85 | 1.00 | 0.85 | 0.92 |
| EM=5 | 92.00% | 84 | 0 | 100 | 16 | 0.00 | 0.84 | 1.00 | 0.84 | 0.91 |
| EM=7 | 90.00% | 81 | 1 | 99 | 19 | 0.01 | 0.81 | 0.99 | 0.81 | 0.89 |
| EM=9 | 92.00% | 84 | 0 | 100 | 16 | 0.00 | 0.84 | 1.00 | 0.84 | 0.91 |
| NN | 98.50% | 97 | 0 | 100 | 3 | 0.00 | 0.97 | 1.00 | 0.97 | 0.98 |

Table 49: wav2vec2-CTC-Ge binary classifiers' goodness-of-fit metrics, with a threshold of maximum 1% FPR (if exist) and a minimum 50% TPR, with 100 benign and 100 Psychoacoustic AEs.

| Classifier | Accuracy | TP | FP | TN | FN | FPR | TPR | Precision | Recall | F1 |
|---|---|---|---|---|---|---|---|---|---|---|
| TD | 94.00% | 91 | 3 | 97 | 9 | 0.03 | 0.91 | 0.97 | 0.91 | 0.94 |
| GC | 94.50% | 90 | 1 | 99 | 10 | 0.01 | 0.90 | 0.99 | 0.90 | 0.94 |
| EM=3 | 85.50% | 74 | 3 | 97 | 26 | 0.03 | 0.74 | 0.96 | 0.74 | 0.84 |
| EM=5 | 92.00% | 84 | 0 | 100 | 16 | 0.00 | 0.84 | 1.00 | 0.84 | 0.91 |
| EM=7 | 83.50% | 68 | 1 | 99 | 32 | 0.01 | 0.68 | 0.99 | 0.68 | 0.80 |
| EM=9 | 90.00% | 80 | 0 | 100 | 20 | 0.00 | 0.80 | 1.00 | 0.80 | 0.89 |
| NN | 97.00% | 94 | 0 | 100 | 6 | 0.00 | 0.94 | 1.00 | 0.94 | 0.97 |

Table 50: wav2vec2-CTC-Ge binary classifiers' goodness-of-fit metrics, with a threshold of maximum 1% FPR (if exist) and a minimum 50% TPR, with 100 noisy data and 100 Psychoacoustic AEs.

| Classifier | Accuracy | TP | FP | TN | FN | FPR | TPR | Precision | Recall | F1 |
|---|---|---|---|---|---|---|---|---|---|---|
| TD | 91.50% | 91 | 8 | 92 | 9 | 0.08 | 0.91 | 0.92 | 0.91 | 0.91 |
| GC | 85.50% | 72 | 1 | 99 | 28 | 0.01 | 0.72 | 0.99 | 0.72 | 0.83 |
| EM=3 | 83.50% | 70 | 3 | 97 | 30 | 0.03 | 0.70 | 0.96 | 0.70 | 0.81 |
| EM=5 | 85.50% | 71 | 0 | 100 | 29 | 0.00 | 0.71 | 1.00 | 0.71 | 0.83 |
| EM=7 | 80.50% | 62 | 1 | 99 | 38 | 0.01 | 0.62 | 0.98 | 0.62 | 0.76 |
| EM=9 | 83.50% | 68 | 1 | 99 | 32 | 0.01 | 0.68 | 0.99 | 0.68 | 0.80 |
| NN | 96.50% | 94 | 1 | 99 | 6 | 0.01 | 0.94 | 0.99 | 0.94 | 0.96 |

**Trf-LAS-CTC-Ma model performance** Tab. 51: C&W AEs vs. benign data, Tab. 52: C&W AEs vs. noisy data, Tab. 53: Psychoacoustic AEs vs. benign data, Tab. 54: Psychoacoustic AEs vs. noisy data.

Table 51: Trf-LAS-CTC-Ma binary classifiers' goodness-of-fit metrics, using a threshold of maximum 1% FPR (if available) and a minimum 50% TPR, with 100 benign data and 100 C&W AEs.

| Classifier | Accuracy | TP | FP | TN | FN | FPR | TPR | Precision | Recall | F1 |
|---|---|---|---|---|---|---|---|---|---|---|
| TD | 96.50% | 94 | 1 | 99 | 6 | 0.01 | 0.94 | 0.99 | 0.94 | 0.96 |
| GC | 98.00% | 97 | 1 | 99 | 3 | 0.01 | 0.97 | 0.99 | 0.97 | 0.98 |
| EM=3 | 96.00% | 93 | 1 | 99 | 7 | 0.01 | 0.93 | 0.99 | 0.93 | 0.96 |
| EM=5 | 96.50% | 93 | 0 | 100 | 7 | 0.00 | 0.93 | 1.00 | 0.93 | 0.96 |
| EM=7 | 96.00% | 92 | 0 | 100 | 8 | 0.00 | 0.92 | 1.00 | 0.92 | 0.96 |
| EM=9 | 96.00% | 92 | 0 | 100 | 8 | 0.00 | 0.92 | 1.00 | 0.92 | 0.96 |
| NN | 95.50% | 92 | 1 | 99 | 8 | 0.01 | 0.92 | 0.99 | 0.92 | 0.95 |

Table 52: Trf-LAS-CTC-Ma binary classifiers' goodness-of-fit metrics, using a threshold of maximum 1% FPR (if applicable) and a minimum 50% TPR, with 100 noisy data and 100 C&W AEs.

| Classifier | Accuracy | TP | FP | TN | FN | FPR | TPR | Precision | Recall | F1 |
|---|---|---|---|---|---|---|---|---|---|---|
| TD | 89.50% | 80 | 1 | 99 | 20 | 0.01 | 0.80 | 0.99 | 0.80 | 0.88 |
| GC | 95.00% | 91 | 1 | 99 | 9 | 0.01 | 0.91 | 0.99 | 0.91 | 0.95 |
| EM=3 | 90.50% | 82 | 1 | 99 | 18 | 0.01 | 0.82 | 0.99 | 0.82 | 0.90 |
| EM=5 | 92.50% | 85 | 0 | 100 | 15 | 0.00 | 0.85 | 1.00 | 0.85 | 0.92 |
| EM=7 | 92.00% | 84 | 0 | 100 | 16 | 0.00 | 0.84 | 1.00 | 0.84 | 0.91 |
| EM=9 | 91.00% | 82 | 0 | 100 | 18 | 0.00 | 0.82 | 1.00 | 0.82 | 0.90 |
| NN | 90.00% | 81 | 1 | 99 | 19 | 0.01 | 0.81 | 0.99 | 0.81 | 0.89 |

Table 53: Trf-LAS-CTC-Ma binary classifiers' goodness-of-fit metrics, with a threshold of maximum 1% FPR (if exist) and a minimum 50% TPR, with 100 benign and 100 Psychoacoustic AEs.

| Classifier | Accuracy | TP | FP | TN | FN | FPR | TPR | Precision | Recall | F1 |
|---|---|---|---|---|---|---|---|---|---|---|
| TD | 96.00% | 93 | 1 | 99 | 7 | 0.01 | 0.93 | 0.99 | 0.93 | 0.96 |
| GC | 98.00% | 97 | 1 | 99 | 3 | 0.01 | 0.97 | 0.99 | 0.97 | 0.98 |
| EM=3 | 95.00% | 91 | 1 | 99 | 9 | 0.01 | 0.91 | 0.99 | 0.91 | 0.95 |
| EM=5 | 96.50% | 93 | 0 | 100 | 7 | 0.00 | 0.93 | 1.00 | 0.93 | 0.96 |
| EM=7 | 96.00% | 92 | 0 | 100 | 8 | 0.00 | 0.92 | 1.00 | 0.92 | 0.96 |
| EM=9 | 95.50% | 91 | 0 | 100 | 9 | 0.00 | 0.91 | 1.00 | 0.91 | 0.95 |
| NN | 95.50% | 91 | 0 | 100 | 9 | 0.00 | 0.91 | 1.00 | 0.91 | 0.95 |

Table 54: Trf-LAS-CTC-Ma binary classifiers' goodness-of-fit metrics, with a threshold of maximum 1% FPR (if exist) and a minimum 50% TPR, with 100 noisy data and 100 Psychoacoustic AEs.

| Classifier | Accuracy | TP | FP | TN | FN | FPR | TPR | Precision | Recall | F1 |
|---|---|---|---|---|---|---|---|---|---|---|
| TD | 88.50% | 78 | 1 | 99 | 22 | 0.01 | 0.78 | 0.99 | 0.78 | 0.87 |
| GC | 95.00% | 91 | 1 | 99 | 9 | 0.01 | 0.91 | 0.99 | 0.91 | 0.95 |
| EM=3 | 88.00% | 77 | 1 | 99 | 23 | 0.01 | 0.77 | 0.99 | 0.77 | 0.87 |
| EM=5 | 90.50% | 81 | 0 | 100 | 19 | 0.00 | 0.81 | 1.00 | 0.81 | 0.90 |
| EM=7 | 90.00% | 80 | 0 | 100 | 20 | 0.00 | 0.80 | 1.00 | 0.80 | 0.89 |
| EM=9 | 88.50% | 77 | 0 | 100 | 23 | 0.00 | 0.77 | 1.00 | 0.77 | 0.87 |
| NN | 91.50% | 84 | 1 | 99 | 16 | 0.01 | 0.84 | 0.99 | 0.84 | 0.91 |

**Trf-LAS-CTC-En model performance** Tab. 55: C&W AEs vs. benign data, Tab. 56: C&W AEs vs. noisy data, Tab. 57: Psychoacoustic AEs vs. benign data, Tab. 58: Psychoacoustic AEs vs. noisy data.

Table 55: Trf-LAS-CTC-En binary classifiers' goodness-of-fit metrics, using a threshold of maximum 1% FPR (if available) and a minimum 50% TPR, with 100 benign data and 100 C&W AEs.

| Classifier | Accuracy | TP | FP | TN | FN | FPR | TPR | Precision | Recall | F1 |
|---|---|---|---|---|---|---|---|---|---|---|
| TD | 93.00% | 87 | 1 | 99 | 13 | 0.01 | 0.87 | 0.99 | 0.87 | 0.93 |
| GC | 99.50% | 100 | 1 | 99 | 0 | 0.01 | 1.00 | 0.99 | 1.00 | 1.00 |
| EM=3 | 100.00% | 100 | 0 | 100 | 0 | 0.00 | 1.00 | 1.00 | 1.00 | 1.00 |
| EM=5 | 100.00% | 100 | 0 | 100 | 0 | 0.00 | 1.00 | 1.00 | 1.00 | 1.00 |
| EM=7 | 100.00% | 100 | 0 | 100 | 0 | 0.00 | 1.00 | 1.00 | 1.00 | 1.00 |
| EM=9 | 100.00% | 100 | 0 | 100 | 0 | 0.00 | 1.00 | 1.00 | 1.00 | 1.00 |
| NN | 100.00% | 100 | 0 | 100 | 0 | 0.00 | 1.00 | 1.00 | 1.00 | 1.00 |

Table 56: Trf-LAS-CTC-En binary classifiers' goodness-of-fit metrics, using a threshold of maximum 1% FPR (if applicable) and a minimum 50% TPR, with 100 noisy data and 100 C&W AEs.

| Classifier | Accuracy | TP | FP | TN | FN | FPR | TPR | Precision | Recall | F1 |
|---|---|---|---|---|---|---|---|---|---|---|
| TD | 83.50% | 73 | 6 | 94 | 27 | 0.06 | 0.73 | 0.92 | 0.73 | 0.82 |
| GC | 93.50% | 88 | 1 | 99 | 12 | 0.01 | 0.88 | 0.99 | 0.88 | 0.93 |
| EM=3 | 97.00% | 94 | 0 | 100 | 6 | 0.00 | 0.94 | 1.00 | 0.94 | 0.97 |
| EM=5 | 97.00% | 94 | 0 | 100 | 6 | 0.00 | 0.94 | 1.00 | 0.94 | 0.97 |
| EM=7 | 97.00% | 94 | 0 | 100 | 6 | 0.00 | 0.94 | 1.00 | 0.94 | 0.97 |
| EM=9 | 97.00% | 94 | 0 | 100 | 6 | 0.00 | 0.94 | 1.00 | 0.94 | 0.97 |
| NN | 96.50% | 93 | 0 | 100 | 7 | 0.00 | 0.93 | 1.00 | 0.93 | 0.96 |

Table 57: Trf-LAS-CTC-En binary classifiers' goodness-of-fit metrics, with a threshold of maximum 1% FPR (if exist) and a minimum 50% TPR, with 100 benign and 100 Psychoacoustic AEs.

| Classifier | Accuracy | TP | FP | TN | FN | FPR | TPR | Precision | Recall | F1 |
|---|---|---|---|---|---|---|---|---|---|---|
| TD | 93.00% | 87 | 1 | 99 | 13 | 0.01 | 0.87 | 0.99 | 0.87 | 0.93 |
| GC | 99.00% | 99 | 1 | 99 | 1 | 0.01 | 0.99 | 0.99 | 0.99 | 0.99 |
| EM=3 | 99.00% | 99 | 1 | 99 | 1 | 0.01 | 0.99 | 0.99 | 0.99 | 0.99 |
| EM=5 | 99.50% | 100 | 1 | 99 | 0 | 0.01 | 1.00 | 0.99 | 1.00 | 1.00 |
| EM=7 | 99.50% | 100 | 1 | 99 | 0 | 0.01 | 1.00 | 0.99 | 1.00 | 1.00 |
| EM=9 | 99.50% | 100 | 1 | 99 | 0 | 0.01 | 1.00 | 0.99 | 1.00 | 1.00 |
| NN | 99.00% | 99 | 1 | 99 | 1 | 0.01 | 0.99 | 0.99 | 0.99 | 0.99 |

Table 58: Trf-LAS-CTC-En binary classifiers' goodness-of-fit metrics, with a threshold of maximum 1% FPR (if exist) and a minimum 50% TPR, with 100 noisy data and 100 Psychoacoustic AEs.

| Classifier | Accuracy | TP | FP | TN | FN | FPR | TPR | Precision | Recall | F1 |
|---|---|---|---|---|---|---|---|---|---|---|
| TD | 84.50% | 75 | 6 | 94 | 25 | 0.06 | 0.75 | 0.93 | 0.75 | 0.83 |
| GC | 91.00% | 83 | 1 | 99 | 17 | 0.01 | 0.83 | 0.99 | 0.83 | 0.90 |
| EM=3 | 91.50% | 84 | 1 | 99 | 16 | 0.01 | 0.84 | 0.99 | 0.84 | 0.91 |
| EM=5 | 91.50% | 84 | 1 | 99 | 16 | 0.01 | 0.84 | 0.99 | 0.84 | 0.91 |
| EM=7 | 92.50% | 86 | 1 | 99 | 14 | 0.01 | 0.86 | 0.99 | 0.86 | 0.92 |
| EM=9 | 92.50% | 86 | 1 | 99 | 14 | 0.01 | 0.86 | 0.99 | 0.86 | 0.92 |
| NN | 94.50% | 90 | 1 | 99 | 10 | 0.01 | 0.90 | 0.99 | 0.90 | 0.94 |

## A.10 WORD SEQUENCE LENGTH IMPACT

In nearly all instances, we noticed no substantial decrease in performance, consistently maintaining an AUROC score exceeding 98% across all models, regardless of the word sequence length, supported by the results given in Table 59.

Table 59: Comparing AUROC scores across various word sequence lengths using a combined dataset of 100 benign samples and 100 C&W AEs.

| Model | 2 | 3 | 4 | 5 | 6 | 7 | 8 | 9 | 10+ |
|---|---|---|---|---|---|---|---|---|---|
| LSTM (It) | - | 1.000 | 1.000 | 1.000 | 1.000 | 1.000 | 1.000 | 1.000 | 0.994 |
| LSTM (En) | 1.000 | - | 1.000 | 1.000 | 1.000 | 1.000 | 1.000 | 1.000 | 0.998 |
| LSTM (En-LM) | 1.000 | | 0.917 | 1.000 | 1.000 | 1.000 | 1.000 | 1.000 | 0.995 |
| wav2vec (Ma) | | - | 1.000 | 1.000 | 0.986 | 1.000 | 0.991 | 1.000 | 1.000 |
| wav2vec (Ge) | - | | 1.000 | 1.000 | 1.000 | 0,992 | 1.000 | 1.000 | 1.000 |
| Trf (Ma) | - | - | 1.000 | 1.000 | 1.000 | 1.000 | 1.000 | 1.000 | 1.000 |
| Trf (En) | 1.000 | - | 1.000 | 1.000 | 1.000 | 1.000 | 1.000 | 1.000 | 1.000 |

## A.11 TRANSFERRED ATTACK

To assess the transferability of targeted adversarial attacks between models, we tested whether the effectiveness of these attacks, specifically tailored to one model, remains consistent when tested on another ASR system. The results indicate a lack of transferability, as the WER in all cases is much closer to 100% than the expected 0%, as depicted in Tab. 60 and Tab. 61.

Table 60: WER performance of transferred attacks on chosen pairs of source and target models.

| Source Model | Target model | C&W | Psychoacoustic |
|---|---|---|---|
| wav2vec (Ma) | Trf (Ma) | 99.33% | 99.24% |
| Trf (Ma) | wav2vec (Ma) | 99.41% | 99.41% |

Table 61: WER performance of transferred attacks on chosen pairs of source and target models.

| Source Model | Target model | C&W | Psychoacoustic |
|---|---|---|---|
| LSTM (En) | LSTM (En-LM) | 103.58% | 103.28% |
| LSTM (En) | Trf (En) | 104.48% | 104.58% |
| LSTM (En-LM) | LSTM (En) | 104.98% | 104.68% |
| LSTM (En-LM) | Trf (En) | 104.68% | 104.68% |
| Trf (En) | LSTM (En) | 106.17% | 105.37% |
| Trf (En) | LSTM (En-LM) | 104.68% | 104.78% |

