# OpenReview forum: "Leveraging characteristics of the output distribution for identifying adversarial audio examples"
_ICLR.cc/2024/Conference — Submitted to ICLR 2024_

### Official Review · Reviewer_DwAh · 2023-10-30

**Soundness:** 3 good
**Presentation:** 3 good
**Contribution:** 3 good
**Rating:** 5
**Confidence:** 4

**Summary:**

This paper proposed a new adversarial example detection method for any automatic speech recognition (ASR) system. Relying on the characteristics of the output distribution in ASR system over the tokens from the output vocabulary, the authors use a function to compute corresponding scores and then employ a binary classifier for adversarial detection. Empirical results have demonstrated the effectiveness of the detection method. In addition, to better analyze the robustness of the proposed detection method, the authors also perform adaptive attacks with aware of the defense mechanism.

**Strengths:**

1. This paper proposes a simple and effective method for detection adversarial examples for ASR systems.
2. The paper is presented with comprehensive experiments. The authors not only present the benign detection performance of adversarial attacks but also analysis the robustness under adaptive attacks with known detection method.

**Weaknesses:**

1. Detection performance is only evaluated on limited adversarial attack methods. The authors only evaluate their method on C&W attack and Psychoacoustic attack. More attack methods like [1] [2] and even some black box methods like FAKEBOB [3] are still needed to be included to prove the general performance of the detection method.
2. Lack of comparison with other audio adversarial example detection methods like [4].

[1] Aleksander Madry, Aleksandar Makelov, Ludwig Schmidt, Dimitris Tsipras, and Adrian Vladu. Towards deep learning models resistant to adversarial attacks. In International Conference on Learning Representations, 2018.

[2] Alzantot, Moustafa, Bharathan Balaji, and Mani Srivastava. Did you hear that? adversarial examples against automatic speech recognition. arXiv preprint arXiv:1801.00554 (2018).

[3] Guangke Chen, Zhe Zhao, Fu Song, Sen Chen, Lingling Fan, Feng Wang, and Jiashui Wang. Towards understanding and mitigating audio adversarial examples for speaker recognition. IEEE Transactions on Dependable and Secure Computing, 2022.

[4] Rajaratnam, Krishan, and Jugal Kalita. Noise flooding for detecting audio adversarial examples against automatic speech recognition. In 2018 IEEE International Symposium on Signal Processing and Information Technology (ISSPIT), pp. 197-201. IEEE, 2018.

**Questions:**

In Section5.1, four guiding principles are provided on selection process. Do such principles limit the adversarial attack implementation? For example, there should be an equal number of tokens in both the original and target transcriptions. Would there be other attack scenarios like tokens insertion or deletion?

**Details Of Ethics Concerns:**

No ethics concerns.

---

> ### Author Response · Authors · 2023-11-22
> **Our response to Reviewer DwAh**
>
> Many thanks for your valuable feedback. We performed further investigations based on your comments and made updates to our paper accordingly.
> \
> \
> Q: Detection performance is only evaluated on limited adversarial attack methods. The authors only evaluate their method on C&W attack and Psychoacoustic attack. More attack methods like [1] [2] and even some black box methods like FAKEBOB [3] are still needed to be included to prove the general performance of the detection method.
>
> A: Thanks for your suggestions, we extended our analysis to three more attacks: PDG and those proposed in [1] and [2]. We evaluated the detection performance of our classifiers across diverse attacks. Our findings reveal that our GCs utilizing mean-median characteristic and NNs exhibited successful transferability to the Psychoacoustic attack. Moreover, when exposed to untargeted attacks, they outperformed the baseline in comparison to PGD and Kenansville. For enhanced clarity, we introduced a new analysis, presented in the results section 5.3 on pages 7-8 and table 6 of our revised manuscript.
>
> We did not include FAKEBOB in our analysis, since it is designed specifically for speaker recognition systems, which falls outside the scope of our research.
> \
> \
> \
> Q: Lack of comparison with other audio adversarial example detection methods like [4].
>
> A: Thanks for your input. We included [4] as a baseline in our analysis, but found that it is less effective than our method. Please refer to section 5.3, page 8, and table 4 in our latest manuscript.
> \
> \
> \
> Q: In Section5.1, four guiding principles are provided on selection process. Do such principles limit the adversarial attack implementation? Would there be other attack scenarios like tokens insertion or deletion?
>
> A: Thank you for expressing your concern. The decision to limit the audio length to up to five seconds, was a strategy compromise, balancing time and resources, as generating longer adversarial examples take much more time.
>
> The decision to choose unique target transcriptions is primarily made to minimize bias in our results, ensuring the inclusion of the broadest possible variety of selected words.
>
> The choice of the number of tokens for constructing a target transcription is influenced by the nature of the duration of an audio. There is always a limit of the number of tokens we can hide in an adversarial example, for shorter inputs, achieving a longer target transcription becomes more challenging, and vice versa. Even when the total number of tokens in both the original and target transcriptions are equal, the transcription length itself may vary, as shown in our Demo: https://confunknown.github.io/characteristics_demo_AEs/
>
> This design choice is also supported by previous research. For example, [5] conducted an evaluation on the rate of phones that could be inserted per second without compromising adversarial attack performance. Their findings indicated that as the phone rate increases, the WER also increases. They observed four phonemes per second as a reasonable choice.
>
> [5] Schönherr et al., Adversarial attacks against automatic speech recognition systems via psychoacoustic hiding. In Network and Distributed System Security Symposium (NDSS), 2019.

---

> > ### Comment · Reviewer_DwAh · 2023-11-23
> >
> > Thanks for your response. I appreciate your efforts on adding more experiments of different attack methods and detection baselines. I have increased my score.

---

### Official Review · Reviewer_NsBd · 2023-10-31

**Soundness:** 2 fair
**Presentation:** 2 fair
**Contribution:** 2 fair
**Rating:** 5
**Confidence:** 4

**Summary:**

This paper addresses the issue of adversarial attacks on automatic speech recognition (ASR) systems, where imperceptible noise can manipulate the output. The authors propose a detection strategy applicable to any ASR system, measuring various characteristics of the output distribution. By employing binary classifiers, including simple threshold-based methods and neural networks, they achieve superior performance in distinguishing adversarial examples from clean and noisy data, with AUROC scores exceeding 99% and 98%, respectively. The method's robustness is tested against adaptive attacks, showcasing its effectiveness in detecting even noisier adversarial clips, preserving the system's robustness.

**Strengths:**

This paper works on important issues and is written clearly.

**Weaknesses:**

The types of attacks considered in this work appear to be limited, as it seems to primarily focus on the C&W attack. Why not consider other attack methods, such as PGD attacks or Transaudio transfer attacks[1]? Being able to defend against transferable adversarial samples would make the paper more practically significant. I appreciate the presentation of the entire work, but the limited consideration of attack types makes it hard for me to be convinced.

If the authors can reasonably address my concerns, I would consider increasing the score accordingly.


[1] G. Qi et al., "Transaudio: Towards the Transferable Adversarial Audio Attack Via Learning Contextualized Perturbations," ICASSP 2023 - 2023 IEEE International Conference on Acoustics, Speech and Signal Processing (ICASSP), Rhodes Island, Greece, 2023, pp. 1-5, doi: 10.1109/ICASSP49357.2023.10096873.

**Questions:**

See above.

---

> ### Author Response · Authors · 2023-11-22
> **Our response to Reviewer NsBd**
>
> Many thanks for your feedback! We run additional experiments with the suggested attacks and updated our paper accordingly.
> \
> \
> Q: Why not consider other attack methods, such as PGD attacks or Transaudio transfer attacks?
>
> A: We expanded our empirical analysis by incorporating additional experiments that explore three untargeted attacks: PGD, Kenansville and Genetic attacks.
> While we made an effort to incorporate the Transaudio attack, constraints on time prevented us from creating an implementation specifically designed for Speechbrain and discussing certain inquiries with the authors. To overcome this, we opted to assess our detectors against two additional black-box attacks—Kenansville and the Genetic—suggested by reviewer NsBd.
> We evaluated the detection performance of our classifiers across diverse attacks. Our findings revealed that our GCs utilizing mean-median characteristic and NNs exhibited successful transferability to the Psychoacoustic attack. Moreover, when exposed to untargeted attacks, they outperformed the baseline in detecting PGD and Kenansville attacks. For enhanced clarity, we introduced a new analysis, presented in the results section 5.3 on pages 7-8, and table 6 of our revised manuscript.

---

### Official Review · Reviewer_jXR8 · 2023-11-05

**Soundness:** 3 good
**Presentation:** 3 good
**Contribution:** 3 good
**Rating:** 5
**Confidence:** 3

**Summary:**

The paper addresses the significant issue of adversarial attacks on automatic speech recognition (ASR) systems, a relevant topic in the field of machine learning and security.  The approach is based on analyzing the probability distribution over output tokens at each time step. This involves examining statistical measures like median, maximum, minimum, entropy, and divergence (KL and JSD). Moreover, the authors claims that their detector is resilience when it comes to dealing with noisy data, meaning they can still effectively detect adversarial attempts even when the audio quality is compromised by noise.

**Strengths:**

- The paper introduces a novel approach to identify adversarial attacks by using statistical characteristics of the output token distributions.

- It has been demonstrated that specific statistical measures, like the mean of the median of probabilities, have an acceptable discriminative capabilities. This implies that the authors have done rigorous empirical analysis to identify which features are most effective.

- The authors mention empirical findings, which suggests that they have tested their approaches on real-world data or experiments, providing evidence for their claims.

**Weaknesses:**

- The proposed defense method relies on statistical features like the mean, median, maximum and minimum extracted from the output token probability distributions over time. While these aggregated metrics can efficiently summarize certain characteristics of the distributions, they may miss more subtle adversarial manipulations. For example, an attack could alter the shape of the distribution while keeping the median relatively unchanged. Or it may flip the probabilities of two unlikely tokens, barely affecting the minimum. So only looking at the summary statistics of the distributions may not be enough to detect all possible manipulations by an adaptive adversary.


- While the proposed approach performs remarkably well empirically, it is mostly relying on simple aggregated features. Exploring more sophisticated methods to represent, compare and analyze

**Questions:**

- The adaptive attacks lower your detection accuracy considerably. Have you looked into ways to make the classifiers more robust? For example, by using adversarial training or adding noise to the features.

- Have you evaluated the computational overhead added by extracting the distributional features and running the classifiers? Is this method efficient enough for real-time usage in production systems?

- You use simple summary statistics to represent the output distributions. What prevents more sophisticated adaptive attacks that preserves these summary statistics but still fools the ASR system?

- Your defense relies on statistical metrics like median and maximum probability diverging for adversarial examples. Have you explored attacks that explicitly optimize to minimize statistical distance from the benign data distribution? This could make the adversarials harder to detect.

- Moreover, can the adversarial optimization problem be formulated to reduce divergence from the benign data distribution, while still fooling the ASR system? What are the challenges in constructing such "distribution-aware" attacks?

---

> ### Author Response · Authors · 2023-11-22
> **Our response to Review jXR8**
>
> We thank for your valuable feedback, we have taken your suggestions into consideration and made updates to our paper accordingly.
> \
> \
> Q: Have you looked into ways to make the classifiers more robust?
>
> A: During the training of all models, to improve generalization, we applied standard data augmentation techniques provided in SpeechBrain: corruption with random samples from a noise collection, removing portions of the audio, dropping frequency bands, and resampling the audio signal at a slightly different rate. We described in the paragraph “ASR system” that is in Section 5.1, Page 5.
> \
> \
> \
> Q: Have you evaluated the computational overhead added by extracting the distributional features and running the classifiers? Is this method efficient enough for real-time usage in production systems?
>
> A: Thanks for asking, we performed experiments to measure the total time the system takes to predict 100 audio clips, utilizing an NVIDIA A40 with a memory capacity of 48 GB. As a result, running the assessment with our detectors took approximately an extra 18.74 ms per sample, therefore the proposed method is suitable for real-time usage. We have incorporated this information into the paper, and provided additional details in Appendix A.3 on page 16.
> \
> \
> \
> Q: Can the optimization problem for adversarial attacks be framed to minimize divergence from the benign data distribution while still deceiving the ASR system? And what challenges exist in developing adaptive attacks that maintain summary statistics while effectively fooling the system?
>
> A: Thanks for the suggestion. This is an interesting idea. Unfortunately, due to the limited amount of time, we did not manage to finalize experiments with the suggested adaptive attack, but we will continue investigating and add the results to the final paper.

---

> > ### Author Response · Authors · 2023-11-23
> > **Our response to Review jXR8**
> >
> > Regarding question
> > \
> > Q: Can the optimization problem for adversarial attacks be framed to minimize divergence from the benign data distribution while still deceiving the ASR system? And what challenges exist in developing adaptive attacks that maintain summary statistics while effectively fooling the system?
> >
> > A: Fortunately, we have successfully completed the experiments with the suggested adaptive attack method. We employed a slightly modified loss function to introduce a penalty, aiming to minimize the statistical divergence from the benign data distribution. For this, we calculated for the same time step the KLD between the output distribution given the benign data $x$ and its adversarial counterpart $\hat x$.
> > This adjustment resulted in improved noise levels with higher SNR values compared to other settings. Nevertheless, detectors remain capable of distinguishing adversarial examples from benign data, as detailed in Table 12 of the appendix A.1 on page 14-15 (refer to setting number 6).

---

### Official Review · Reviewer_2Aka · 2023-11-10

**Soundness:** 2 fair
**Presentation:** 3 good
**Contribution:** 2 fair
**Rating:** 5
**Confidence:** 3

**Summary:**

This paper introduces a method to detect adversarial audio example by exploiting statistical features. Based on the selected features, accurate predictions can be made to differentiate adversarial audio examples and standard audio samples. An adaptive attack against proposed detection methods is also introduced, even though less effective against adversarial examples, the authors claim that the noise level of the adaptive attack is higher and the adaptive adversarial audio examples can be easily picked by the human ear.

**Strengths:**

The paper is very well presented and easy to follow. Extensive experimental results are provided to support the claims. The results demonstrate that the proposed detection is more generally more accurate than the existing TD detection method. Results on adaptive attacks also show that if an adaptive adversarial audio example targets the proposed detection, more audible noises will be included in the adversarial example.

**Weaknesses:**

- The reason behind the selected statistics can be further motivated. Why are these statistical features selected? A related question is why the generated adaptive adversarial audio examples are noisier when optimizing with respect to relevant feature?
- Regarding the generalization of the proposed detection, the transferability of the detection can be further clarified. About intra-model generalization, will the detection model that is trained on one specific kind of adversarial example be generalizable to other types of adversarial examples? This point needs to be clarified since it may weaken the threat model that the detector needs to know the type of the adversarial attack beforehand. About inter-model generalization, will a detector trained on one ASR model be able to detect adversarial examples that are generated on a different ASR model? It would be great if the authors can clarify the generalization of the proposed method.
- About the adaptive attack, have the authors considered other types of attacks that may decrease the noise level of the adversarial audio examples?  I really appreciate that the authors provide experiments on adaptive attacks, which definitely makes the claims stronger. It would be great if the authors could clarify the specific efforts that have been made to control the noise level.

**Questions:**

See questions in the weaknesses.

---

> ### Author Response · Authors · 2023-11-22
> **Our response to Review 2Aka**
>
> We value your insightful comments and queries. To address your concerns, we provide explanations in the following and updated the paper accordingly.
> \
> \
> Q: Why are these statistical features selected?
>
> A: Previous work indicated the high potential of statistics of the output distribution that implement uncertainty measures: on the one hand, the mean entropy was turned into a defense method against adversarial attacks on hybrid ARS system with limited vocabulary, as discussed in [1].  On the other hand, the Kullback-Leibler divergence between the distributions in consecutive steps was used to assess the reliability of ARS systems, see [2]. We therefore aimed to investigate if this also applies for state-of-the-art end-to-end ARS systems. We expanded our analysis to simple characteristics (like max, min, median) and were surprised ourselves that they lead to sometimes even better detection results.
>
> [1] Däubener et al., Detecting adversarial examples for speech recognition via uncertainty quantification. In Proc. Interspeech 2020, pp.4661–4665, 2020.
>
> [2] Meyer et al., Performance monitoring for automatic speech recognition in noisy multi-channel environments. In 2016 IEEE Spoken Language Technology Workshop (SLT), pp. 50–56, 2016.
> \
> \
> \
> Q: Why the generated adaptive adversarial audio examples are noisier when optimizing with respect to relevant feature?
>
> A: The adaptive attack contains two (potentially competing) loss functions, which makes the optimization problem harder. We also were successful in generating less noisy examples with the adaptive attack (see Appendix A.1, page 14) which however were less often fooling the detection model. We added a sentence to make this clearer in the main paper.
> \
> \
> \
> Q: Will be the detection model that is trained on one specific kind of adversarial example be generalizable to other types of adversarial examples?
>
> A: Yes. First, note that the construction of Gaussian classifiers can be done only with benign examples. While we also investigated the performance based on the best characteristic picked on a validation set, we also found that some characteristics perform very well for different attack types and thus can be chosen without any adversarial data. Only benign data for picking the threshold is needed. Moreover, we conducted experiments on intra-model generalization for the neural networks, investigating if a detection model trained on the C&W attack can be applied as a defense method against the other targeted (i.e., Psychoacoustic) and three newly added untargeted attacks. We found that our GCs based on the mean-median characteristic and NNs demonstrated effective transferability to the Psychoacoustic attack. Additionally, when subjected to untargeted attacks, they performed better than the baseline against PGD and Kenansville attacks. To make this more clear, we incorporated a new analysis displayed by the results in section 5.3/pages 7-8 /Table 6 of our updated manuscript.
> \
> \
> \
> Q: Will a detector trained on one ASR model be able to detect adversarial examples that are generated on a different ASR model?
>
> A: We don't expect our detectors to generalize between models, since the statistics for different models differ. Previous research has demonstrated almost no transferability of adversarial examples between speech models, as indicated in [1]. This lack of transferability, especially in the context of targeted optimization attacks, suggests that adversarial examples crafted to target a specific model lose their efficacy when applied to a new ASR model, posing a limited threat. We verified this claim with our models and obtained similar results, with almost no transferability observed between models. These results are reported in Appendix A.11 on pages 30-31.
>
> [1] Abdullah et al., Sok: The faults in our ASRs: An overview of attacks against automatic speech recognition and speaker identification systems. 2021 IEEE Symposium on Security and Privacy (SP), pp. 730–747, 2020.
> \
> \
> \
> Q: About the adaptive attack, have the authors considered other types of attacks that may decrease the noise level of the adversarial audio examples?
>
> A: Yes, we did. As described in details in Appendix A.1 (page 14) we experimented with several different settings that resulted in lower noise levels but also in less affective attacks. We added a sentence to make this clear in the main paper.

---

> ### Comment · Reviewer_2Aka · 2023-12-03
> **Concerns about adaptive attack and explanations on features**
>
> Thank the authors for the detailed clarifications. I appreciate it. However, I will still maintain my original rating due to the limited effectiveness of the proposed approach against adaptive defenses and the limited explanation of the working mechanism behind the approach. My suggestions would be to explain why the selected features work effectively, especially compared with the end-to-end detector, or show the potential of the proposed method against adaptive attacks under a certain bound.

---

### Author Response · Authors · 2023-11-22
**General Response**

We appreciate the reviewers for their insightful feedback. We are thankful the reviewers recognized the clarity of our motivation and idea (R: 2Aka and NsBd) and the novelty in our approach (R: jXR8). Their acknowledgment of the rigor of our experiments and analyses (R: jXR8, 2Aka, DwAh) and the comparison to a suitable baseline (R: 2Aka) is delighting. We're also happy with the recognition that our work addresses a significant issue in machine learning and security (R: NsBd and jXR8).
In response to their input, we substantially expanded our empirical analysis by incorporating additional experiments that explore:
\
\
1- the performance of our defense model with respect to three untargeted attacks, namely the wight-box attack PGD (as suggested by reviewer NsBd and DwAh), and two black box-attacks—the Genetic attack (proposed by reviewer DwAh) and Kenansville attack. While the detection performance for untargeted attacks, which are less harmful but more challenging to detect in general, is not as good as for targeted attacks, our approach is clearly superior compared to previously proposed detection methods (see section 5.3, Table 6 on page 9 of our updated manuscript).

2- the generalization ability of our neural network classifiers, i.e., we tested if the neural networks trained on one kind of adversarial examples are able to detect other kind of adversarial examples. We found great transferability between targeted attacks (refer to section 5.3/Table 6 of our updated manuscript).

3- the effectiveness of an additional adversarial example detection technique, employed as an additional baseline—specifically, Noise Flooding (NF), as recommended by reviewer DwAh. Our results demonstrate that the proposed binary classifiers consistently outperform NF across all models, see section 5.3 and Table 4 of our updated manuscript.

4- the computational overhead added by incorporating a detector into the ASR pipeline (as suggested by reviewer jXR8). We found that running the assessment with our detectors took approximately an extra of 18.74 ms per sample, utilizing an NVIDIA A40 with a memory capacity of 48 GB. (See Appendix A.3 on page 16 of our updated manuscript).
\
\
Please do not hesitate to get in touch with us for any additional comments you may have.

---

> ### Author Response · Authors · 2023-11-23
> **General Response**
>
> Including an additional empirical analysis to the list:
>
> 5- the detectability of an adaptive attack with slightly different adversarial loss function, incorporating a penalty to minimize the statistical distance from the benign data distribution (as recommended by jXR8), see Appendix A.1 and Table 12 on page 14 of our revised manuscript.

---

### Meta-Review · Area_Chair_DVr7 · 2023-12-03

**Metareview:**

The paper proposes a method to detect adversarial attacks of ASR systems. The idea is to train a binary classifier on features of the distribution of the tokens recovered by the ASR system in the presence or absence of an adversarial attack.

One main weakness was that the method was experimentally evaluated against only one attacker (Carlini & Wagner, 2018) but the authors have extended their study to three other attacks in the discussion period.

The principle of comparing output distributions for benign and attacked inputs is an interesting one. But the datasets used to train this classifier can trivialize this detection problem. For example, the librispeech dataset is very specific, and the token frequencies are undoubtedly very particular: the sentence obtained after attack is undoubtedly outside the domain of clean sentences. The authors should have compared attack detection performance on text alone, or when the text produced is part of possible outputs or already observed. For example, does training a generative model of healthy texts and calculating the likelihood of a text produced during an attack give the same results?

Without this kind of evaluation, the significance of the contribution is difficult to estimate.

**Justification For Why Not Higher Score:**

Important evaluations are still missing to reach the ICLR level of acceptance.

**Justification For Why Not Lower Score:**

N/A

---

### Decision · Program_Chairs · 2024-01-16

Reject